# Mechanistic insights into the three steps of poly(ADP-ribosylation) reversal

Johannes Gregor Matthias Rack [1,5], Qiang Liu[2,5], Valentina Zorzini[1], Jim Voorneveld[2], Antonio Ariza[1], Kourosh Honarmand Ebrahimi[3], Julia M. Reber[4], Sarah C. Krassnig [4], Dragana Ahel[1], Gijsbert A. van der Marel[2], Aswin Mangerich [4], James S. O. McCullagh [3], Dmitri V. Filippov [2✉] & Ivan Ahel [1✉]

Poly(ADP-ribosyl)ation (PAR) is a versatile and complex posttranslational modification composed of repeating units of ADP-ribose arranged into linear or branched polymers. This scaffold is linked to the regulation of many of cellular processes including the DNA damage response, alteration of chromatin structure and Wnt signalling. Despite decades of research, the principles and mechanisms underlying all steps of PAR removal remain actively studied. In this work, we synthesise well-defined PAR branch point molecules and demonstrate that PARG, but not ARH3, can resolve this distinct PAR architecture. Structural analysis of ARH3 in complex with dimeric ADP-ribose as well as an ADP-ribosylated peptide reveal the molecular basis for the hydrolysis of linear and terminal ADP-ribose linkages. We find that ARH3-dependent hydrolysis requires both rearrangement of a catalytic glutamate and induction of an unusual, square-pyramidal magnesium coordination geometry.

[1] Sir William Dunn School of Pathology, University of Oxford, Oxford, UK. [2] Leiden University, Leiden Institute of Chemistry, Leiden, The Netherlands. [3] Department of Chemistry, University of Oxford, Chemistry Research Laboratory, Oxford, UK. [4] Molecular Toxicology Group, Department of Biology, University of Konstanz, Konstanz, Germany. [5] These authors contributed equally: Johannes Gregor Matthias Rack, Qiang Liu. ✉email: filippov@chem.leidenuniv.nl; ivan.ahel@path.ox.ac.uk

Poly(ADP-ribose) (PAR) is a polymer of ADP-ribose (ADPr) moieties, established through initial transfer of a single ADPr from β-NAD$^+$ onto a protein acceptor site (termed mono(ADP-ribosyl)ation [MARylation]) followed by polymer extension through repeated conjugation (termed poly(ADP-ribosyl)ation [PARylation]). The different modification variants regulate a wide array of cellular processes including development, transcription, and the DNA damage response (DDR)[1,2]. Three main criteria have been established to influence functional outcomes: the conjugation site, the chain length, and the branch frequency. First, modification of the so far identified acceptor residues (including Asp/Glu, Ser, Tyr, Arg and Cys) results in O-, N- and S-glycosidic linkages, which allows target selection as well as specific regulation of signal turnover[1,3,4]. Serine is the main acceptor in the DDR and its modification is synthesised by PARP1:HPF1 and PARP2:HPF1 complexes[5–8]. Second, the polymer length may vary between 2 and 200 units, thus influencing the number of proteins able to interact with the modification. Furthermore, the charged nature of large PAR polymers can trigger condensate formation, thereby dramatically altering the physicochemical microenvironment[7]. Finally, in addition to linear ribose(1″→2′)ribose linkages, infrequent addition of ribose(1″→2″)ribose(1″→2′)ribose conjugates leads to polymer branching and increase in signal complexity[4,9]. Together, this modification heterogeneity was proposed to form the PAR code dictating the outcome of ADP-ribosylation signalling[2,5]. Consequently, the degradation of PAR can be broken down into three reactions: (i) degradation of the linear ribose(1″→2′)ribose linkage, (ii) hydrolysis of the branch point ribose(1″→2″)ribose linkage and (iii) cleavage of the terminal amino acyl-ADP-ribosyl bond.

Poly(ADP-ribose)glycohydrolase (PARG) and (ADP-ribosyl) hydrolase 3 (ARH3) are the main hydrolases of serine PAR- and MARylation, respectively, thus playing a pivotal role in signal silencing following DNA damage[10]. PARG is responsible for the majority of polymer degradation[11,12], but it is unable to remove the terminal ADP-ribose moiety from the protein substrate[13], whereas ARH3 primarily cleaves the seryl-ADP-ribosyl linkage, and hence terminates the signal[12]. ARH3 is the only known enzyme to reverse serine MARylation and is consequently responsible for the regulation of hundreds of ADP-ribosylated proteins following DNA damage[5,14,15]. In addition, ARH3 possesses weak PAR cleavage activity[16] and it was suggested that this compensates for slower processing of short chains by PARG[10,17,18], or regulates PAR-induced apoptosis (Parthanatos) through degradation of free PAR chains[19]. ARH3 deficiency leads to persistence of MARylation marks on serine residues that influences the local histone modification pattern and was proposed to alter the transcriptional activity at the ADP-ribosylated loci[20,21]. Genetic deficiency in ARH3 has been recently shown to cause stress-induced childhood-onset neurodegeneration with variable ataxia and seizures (CONDSIAS), an autosomal recessive neurodegenerative disorder that in severe cases can lead to childhood death[21–23].

Evolutionarily, PARG and ARH3 belong to unrelated protein families, macrodomains (Pfam CL0223) and (ADP-ribosyl)hydrolases (Pfam PF03747), respectively[24]. Extensive research shows that PARG cleaves PAR preferably from the end of the polymer (exo) via a glutamate-mediated activation of the O-glycosidic bond and formation of an oxocarbenium intermediate[10,17,25]. PARG was also shown to be able to cleave within polymers (endo) and degrade branch points, yet the details of these activities remain elusive[17,26]. In contrast to PARG, the active site of ARH3 contains a catalytic binuclear magnesium centre[11]. Recent structural studies of ARH3 using the reaction product ADP-ribose and its analogues suggest that a crucial aspect aiding substrate turnover is the exact positioning of the substrate within the active site[27–29]. However, these studies showed subtle differences in ligand positioning and provide only limited information of the pre-catalytic substrate binding mode, which led to the proposal of several conflicting mechanisms[27–29]. Furthermore, these studies could not explain the stark differences in turnover rates for the chemically similar linkages in serine-MAR and PAR and thus hamper our understanding of the physiological role of ARH3 in the DDR.

In order to fully understand the PAR reversal reactions, we developed a synthetic route to generate well-defined branched PAR molecules. Using synthetic linear and branched PAR we demonstrate that PARG, but not ARH3, can efficiently reverse PARylation. We provide the molecular basis for these differences by solving the three-dimensional X-ray crystal structures of ARH3 in complex with a MARylated peptide and dimeric ADP-ribose. Our structural data revealed that substrate binding induces structural changes leading to repositioning of a catalytic glutamate residue as well as substrate-assisted alteration of the magnesium coordination geometry. Yet PAR binding fails to efficiently induce these structural in ARH3, thus providing a structural explanation for the observed low hydrolytic activity towards this substrate. Together, our findings clarify the hydrolysis mechanism of ARH3 and help to explain observed differences in its physiological activity.

## Results

**Synthesis of PAR branch point**. Recent studies suggest that the PAR branching frequency is an important determinant of the cellular outcomes of ADP-ribosyl signalling[30,31]. However, the influence of branching on polymer stability remains elusive. To gain insights, we synthesised linear, dimeric and trimeric PAR as reported earlier[32] and describe here a strategy for the synthesis of the PAR chain branch points (Fig. 1). The target branched ADPr-trimer (**1**) has multiple anionic pyrophosphates and was constructed via a solid-phase approach to circumvent repeated isolation of charged intermediates and increase synthesis efficiency. The synthetic route to target compound **1** involved three challenges that needed to be overcome: (i) the installation of three acid-sensitive and anionic pyrophosphate linkages, (ii) the assembly of the tri-riboside core having two 1,2-cis α-glycosidic bonds and (iii) selection of a solid support compatible with the projected chemistries. The introduction of pyrophosphates in oligomer **1** relies on our P(V)-P(III) method for the solution synthesis of sugar nucleotides[33] that was successfully applied to the preparation of linear ADPr oligomers[32]. To adapt the P(V)-P(III) method to the synthesis of branched ADPr-oligomer **1**, we designed advanced branched phosphoramidite building block **3** (Fig. 1). This key building block bears a protective group pattern that allows for orthogonal cleavage of the temporary phosphate protection by treatment with base to exclude acid-catalysed degradation of both the pyrophosphates and the O-glycosidic linkages[32]. We utilised the base-labile fluorenylmethyl (Fm) protecting group, which is amenable to pyrophosphate introduction in combination with known building block **4**[34–36]. Tentagel resin **2** (Fig. 1) was chosen as a solid support due to its nearly universal solvent compatibility and relatively high loading capacity, enabling a straightforward pathway for future scale-up.

Figure 2 depicts the synthesis of parotriose **14**, an intermediate towards key building block **3**. Firstly, D-ribose was allylated, per-methoxybenzylated and finally de-allylated using PdCl$_2$ as a catalyst[37,38], to yield 2,3,5-tri-O-p-methoxylbenzyl-D-ribofuranose **5**. Conversion of **5** into the corresponding imidate donor **6** with 2,2,2-trifluoro-N-phenylacetimidoyl chloride and Cs$_2$CO$_3$ in acetone proceeded in good yield. Gratifyingly, the first TMSOTf mediated glycosylation using donor **6** and known acceptor **7**[32]

**Fig. 1 Retrosynthetic analysis for branched ADPr-trimer (1).** ADPr-trimer **1** with substructures originating from the three distinct building blocks: ribofuranoside **2** immobilized on TentaGel (purple), phosphoramidite of branch point **3** (blue) and adenosine phosphoramidite **4** (orange). The dotted box shows the chemical structure of Fm (fluorenylmethyl group) and *i*Bu (isobutyryl group). Purple sphere represents TentaGel resin. Bz benzoyl, Ac acetyl, Q-linker hydroquinone-*O,O′*-diacetyl.

**Fig. 2 Synthesis of protected parotriose (14).** D-Ribose is the starting material for the synthesis of donor **6** which is further coupled with acceptor **7** and donor **11** to obtain parotriose derivative **14**. Yield after each step is given as percentage in brackets. PMB *para*-methoxylbenzyl, TMSOTf trimethylsilyl trifluoromethanesulfonate, DCM dichloromethane, TIPDS 1,1,3,3-tetraisopropyldisiloxane, TIPS triisopropylsilyl, Bn benzyl, TEA-HF Triethylamine trihydrofluoride, TBDPS *tert*-butyldiphenylsilyl.

furnished solely α-configured disaccharide **8** in high yield. Attempted deprotection of the PMBs with DDQ resulted in a low yield due to the formation of a 2,3-methoxybenzylidene side product. However, acidolysis using TFA rapidly cleaved off all PMB groups, affording triol **9** in reasonable yields. Next, the 3- and 5-OH in **9** were silylated to give TIPDS (1,1,3,3-tetraisopropyldisiloxane)

protected **10**, allowing selective glycosylation on 2-OH. Condensation of donor **6** and acceptor **10** using TMSOTf followed by subsequent deprotection of the PMB groups did yield the desired trisaccharide but in a low yield (24% from **10**, see Supplementary Information) which can be attributed to acidic cleavage of one or both glycosidic bonds during PMB deprotection. To avoid the

**Fig. 3 Synthesis of branch point phosphoramidite (3).** Parotriose derivative **14** is converted into nucleotide **15** via a Vorbrüggen type glycosylation after which phosphate and phosphoramidite functionalities are installed at the primary hydroxyls of the ribose residues. Red: the arrows show the notation for the three distinct 5-OHs. Yield after each step is given as percentage in brackets. HClO₄-SiO₂ perchloric acid on silica gel, BSTFA N,O-bis(trimethylsilyl) trifluoroacetamide, DMT 4,4′-dimethoxytrityl, TBAF tetra-n-butylammonium fluoride, DCI 4,5-dicyanoimidazole, tBuOOH tert-butyl hydroperoxide, TFA triflouroacetic acid, DIPEA N,N-Diisopropylethylamine, DMF dimethylformamide.

problematic acidolysis, we coupled acceptor **10** with a different donor **11**, bearing Bn protecting groups on the 2- and 3-OH instead, using the same TMSOTf/DCM condition to give trisaccharide **12** in good yield and with excellent α stereoselectivity. Pd/C catalysed high-pressure (80 bar) hydrogenolysis of the Bn groups was followed by TEA·3HF mediated desilylation to produce **13** in 79% yield. Selective silylation of both primary hydroxyls in pentol **13** followed by acetylation of all secondary OH-groups to furnish protected parotriose **14** in 82% yield.

The synthesis continued with introduction of $N^6$-benzoyl adenine at the anomeric position of **14** via a Vorbrüggen type glycosylation catalysed by HClO₄-SiO₂ to afford protected parotriosyl adenine **15** (Fig. 3)[39]. To make **15** ready for introduction of the two phosphotriesters on the primary 5″- and 5‴-OH positions, all acetyl and benzoyl esters were carefully saponified with 1 M aqueous NaOH at 0 °C to give **16** in a high yield. It is well-established that the N-benzoyl at the exocyclic amino group of adenine remains stable under these alkaline conditions[32,39–41]. In order to ensure the orthogonality between the three primary hydroxyls in **16**, the released 5′-OH was protected with DMT (4,4′-Dimethoxytrityl) and the remaining secondary alcohols were acetylated in the same reaction vessel to give **17**. The two silyl groups in **17** were then carefully removed by TBAF to liberate the terminal 5″-OH and 5‴-OH (**18**), allowing access to the following, high yielding and single-operation reaction cascade to **20**. The first step in this reaction sequence consisted of the DCI-catalysed phosphitylation of the 5″- and 5‴ OH-groups with Fm amidite **19**[42,43] followed by oxidation of the resulting phosphites to phosphate triesters by tBuOOH. Next, stoichiometric amounts of TFA were used to rapidly remove the DMT leading to compound **20** with deprotected 5-OH′. Finally, treatment of alcohol **20** with commercially available aminophosphorochloridite **21** and DIPEA in DMF gave key phosphoramidite **3**. It is important to note that this phosphitylation needs a careful work-up procedure for the removal of DIPEA, as DIPEA is capable of cleaving one of the Fm groups. In addition, the simultaneous occurrence of an acid labile (phosphoramidite) and the base-labile (Fm) groups in compound **3** requires column chromatography with high-quality IRR silica gel (see Supplementary Information). Eventually, starting from D-ribose, the advanced phosphoramidite **3** was prepared via a 21-steps high yielding, highly stereoselective synthetic route in

sufficient amount (0.36 mmol) for the purpose of the solid-phase synthesis of branched ADPr oligomers.

After preparation of all required building blocks (see Supplementary Methods for preparation of functionalised Tentagel resin **2**), we proceeded with the solid-phase synthesis of branched ADPr-trimer **1** in an iterative P(V)-P(III) procedure (Fig. 4). An α-O-methyl group was installed at the anomeric centre at the terminal ribose in immobilized **2** to mimic the native PAR stereochemistry. DBU (10%)-mediated removal of Fm protections on resin **2** was followed by the introduction of the first pyrophosphate function by a three-step procedure: (i) 5-(benzylthio)-1H-tetrazole (BTT) assisted coupling of advanced phosphoramidite building block **3** with the deprotected phosphate derivative of **2**, (ii) CSO oxidation of the resulting phosphite–phosphate intermediate and (iii) DBU mediated cleavage of both CE and terminal Fm groups. The obtained immobilized intermediate **22**, with two phosphomonoester functions allows the simultaneous introduction of the next two pyrophosphates by the same three-step P(V)-P(III) coupling cycle using phosphoramidite **4**. Finally, removing all the protecting groups and the cleavage of the product from the resin using aqueous NH₄OH gave the desired branched ADPr-trimer **1**. Purification with anion-exchange chromatography led to the isolation of 0.68 mg of target branched ADPr **1**. The side product of the reaction, α-1″-O-methyl-ADP-ribose **23** (meADPr), was recovered separately (Fig. 4 and Supplementary Table 1).

## PARG and ARH3 cooperate in PAR reversal. Following synthesis, we first tested the efficiency of PARG and human ARH3 (hARH3) to initiate branch point reversal by measuring their ability to degrade the synthetic PAR trimers. We used thin-layer chromatography (TLC) and liquid chromatography high-resolution mass spectrometry (LC-HRMS) (see Methods) and found that PARG was able to efficiently degrade the branch point, whereas hARH3 showed no activity under these assay conditions (Fig. 5a, b and Supplementary Figs. 1a, 2). Long time-course experiments revealed that hARH3 is able to degrade branched trimers, however, this activity is dramatically lower than the cleavage of the linear polymer and, hence, may not be physiologically significant (Fig. 5c and Supplementary Fig. 1b). This is further supported by our observations in cells: we cultured U2OS wild-type (wt) and ARH3⁻/⁻ cells and analysed cellular content

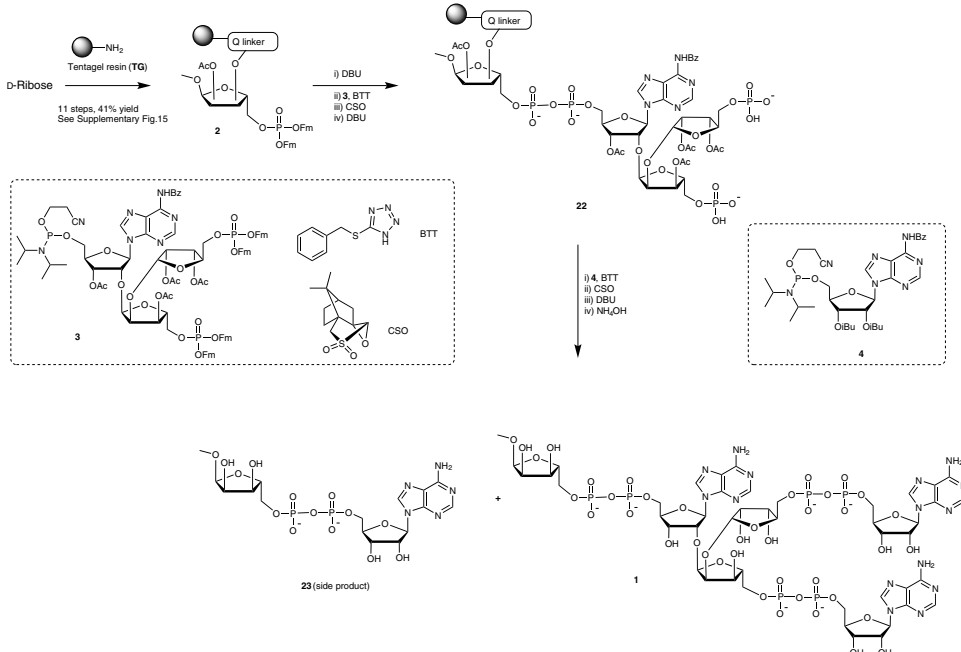

**Fig. 4 Solid-phase synthesis of branched ADPr-trimer (1).** Riboside immobilized on TentaGel resin **2** is synthesized from D-Ribose in eleven steps, then phosphoramidites **3** and **4** are coupled sequentially to furnish branched trimer **1**. Monomer **23** is found as a side product. Yield after each step is given as percentage in brackets. The two dotted boxes show the chemical structures of the building blocks (phosphoramidite **3** and **4**) and reagents (BTT, CSO) used in these reactions. DBU 1,8-diazabicyclo(5.4.0)undec-7-ene, BTT 5-benzylthio-1*H*-tetrazole, CSO (1*S*)-(+)-(10-camphorsulfonyl)oxaziridine, NH$_4$OH ammonia hydroxide.

of linear and branched PAR through isolation of di- to mono-ribosyl-adenosine (R$_2$-Ado/R-Ado; nucleoside utilised as indicators of branching and linear PAR, respectively) as described earlier[44,45]. In the absence of PARG inhibitor (PARGi), the levels of R$_2$-Ado were below the technical limit of detection. However, they could be readily detected in PARG-inhibited cells, in line with PARG's involvement in branching removal (Fig. 5d). Interestingly, we observed a ~1.5-fold lower branch frequency (R$_2$-Ado/R-Ado) in PARGi-treated ARH3$^{-/-}$ cells compared to PARGi-treated wt cells. These data suggest that ARH3 preferentially degrades linear PAR chains and argue against the importance of ARH3 for branch point removal in vivo.

Interestingly, LC-HRMS analysis of the PARG catalytically impaired mutant E756N showed marginal activity against the branch point, which allowed us to detect a meADPr-ADPr dimer as a reaction intermediate, thus identifying the cleavage of the ribose(1″→2″)ribose bond as the first step in branch point resolution (Fig. 5b and Supplementary Fig. 2). This finding is in line with earlier observations suggesting preferential branch point cleavage by PARG[26,46]. Previous structural and functional studies showed that efficient hydrolysis requires the formation of a substrate:PARG complex involving coordination of the 2″-OH group by Glu755 and Glu756 and the 3″-OH moiety by N740[13,47,48]. To gain insight into the PARG:branch point interaction, we produced energy minimised models of PARG with branched ADPr as ligand (Supplementary Fig. 3). Analysis of the ligand coordination showed that placement of ADPr$_{a2:1}$ (linear polymer extension; see Supplementary Note 1 for proposed PAR nomenclature) within the active site prevents E756 from coordinating 2″O (scissile bond oxygen) due to steric hindrance imposed by the [ADPr$_{b1:1}$] distal ribose (Supplementary Fig. 3), thus preventing hydrolysis. Furthermore the 2″ and 3″OH moieties of [ADPr$_{b1:1}$] may have a slight shielding effect towards the 3″OH of ADPr$_{a2:1}$ influencing its interaction with N740. Contrarily, placement of [ADPr$_{b1:1}$] within the active site

does not impose these constraints on the pre-catalytic complex and a coordination comparable to that observed in PARG:dimer structure (PDB 5A7R) appears feasible (Supplementary Fig. 3). While further work is needed, our data argue for a PAR degradation mechanism in which branch pruning is required prior to polymer degradation beyond the branch point.

To assess whether the 1″-O-methyl moiety of ADPr$_{a1:n}$ had an influence on the reaction, we performed inhibitor studies utilising free meADPr. Our results indicate that the latter acts as an inhibitor of the ARH3 reaction with comparable potency to ADPr (Supplementary Fig. 4a). To assess whether these findings are transferable to enzyme-derived PAR, we generated different serine-linked PAR polymers formed by PARP1 variants with defined influence on polymer formation: G972R (hypo-branching), Y986H (hyper-branching) and Y986S (short chain, wt branch frequency)[30,49] in the presence of HPF1[5,7]. Indeed, the stability of PARP1 Y986H-derived polymers was increased in comparison to G972R- or Y986S-derived PAR (Fig. 5e), thus further supporting our finding that ARH3 hydrolysis of PAR is slowed down by the presence of branch points.

**Structures of ARH3 bound to its substrates.** Amongst the known (ADP-ribosyl)hydrolases—both of the macrodomain and ARH superfamilies—ARH3 stands out due to its ability to cleave diverse chemical linkages including the acetal O-glycosidic bond found in PAR and serine MARylation (Supplementary Figs. 4b, 5a and Table 1), the ester O-glycosidic bond of the metabolic intermediates 1″-O-acetyl-ADP-ribose (OAADPr)[16,50] as well as the N-glycosidic bond found in α-NAD$^+$ (Supplementary Fig. 5b and Supplementary Table 1)[51]. Yet, the available ARH3:product and product analogue structures did not provide insights as to why chemically similar bonds are cleaved with different efficiency: ARH3 rapidly cleaves the seryl-ADP-ribosyl bond, but PAR degradation is slow. OAADPr is readily converted, but aspartate/glutamate MARylation hydrolysis is negligible[12]. Lastly, α-NAD$^+$

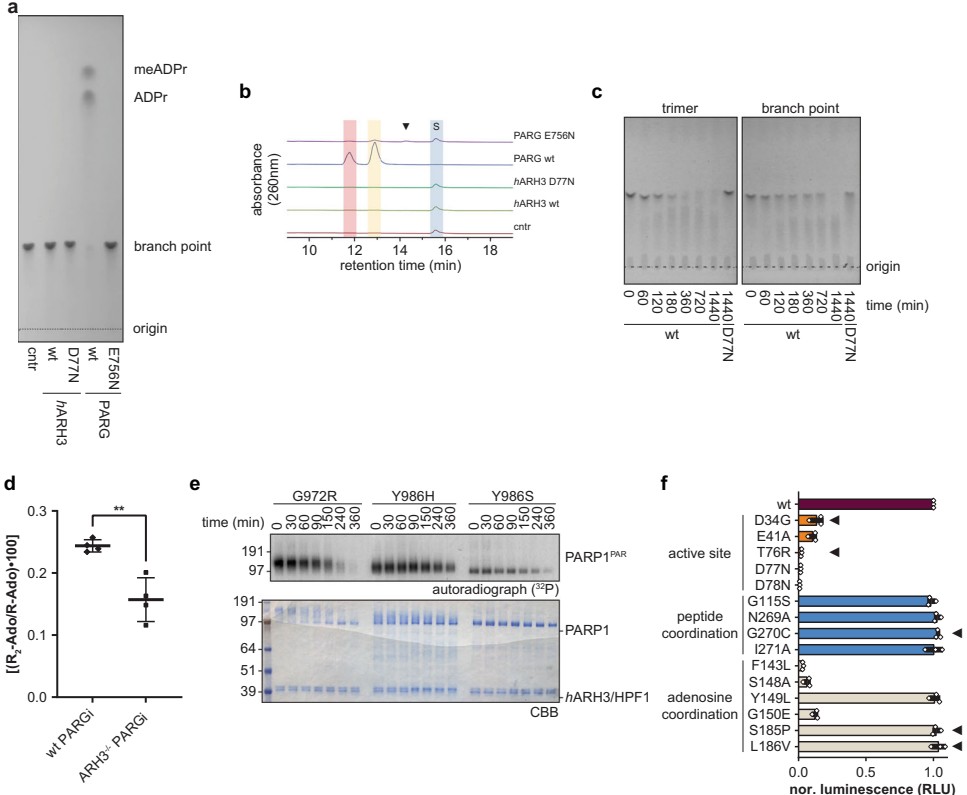

**Fig. 5 PAR branching impairs the glycohydrolase activity of ARH3. a** Synthetic PAR trimer degradation by ARH3 and PARG. Samples were analysed by thin-layer chromatography (TLC) and representative results of three independent experiments are shown. **b** UV-visible chromatograms of LC-HRMS analysis of samples represented in (**a**). S, PAR branch point and black triangle, PARG E756N specific peak. Full analysis is shown in Supplementary Fig. 2. **c** Time-course of synthetic PAR trimer and branch point degradation by hARH3. Samples were analysed by TLC and representative results of two independent experiments are shown. **d** PAR branching levels in wt or ARH3$^{-/-}$ U2OS cells. Wt and ARH3$^{-/-}$ U2OS cells were treated with 25 µM PARGi for 4 days. Afterwards, the di-ribosyl-adenosine to ribosyl-adenosine (R$_2$-Ado/R-Ado) ratio was quantified by UPLC-MS/MS analysis (see 'Methods'). Data are shown as mean ± s.d. of biological quadruplicates, **$p < 0.0032$ (unpaired, two-tailed $t$-test). **e** Time-course of hydrolysis of enzymatic Ser-PARylation by hARH3. To achieve Ser-PARylation PARP1 mutants were modified with $^{32}$P-NAD$^+$ in presence of HPF1. Shown are representative results of three independent experiments. **f** Hydrolysis of Ser-MARylation by hARH3 wt and mutants using H2BS7mar peptide as substrate. Briefly, the released ADPr was converted by NUDT5 to AMP, which in turn was detected by chemical luminescence using the AMP-Glo™ assay (Promega; see 'Methods'). Samples are background corrected and normalised to wt. Data are presented as mean values ± s.e.m. from three independent replicates measured in triplicates. COSMIC database-derived mutations are indicated by black triangle. Samples were analysed by SDS-PAGE followed by Coomassie Brilliant Blue (CBB) staining and autoradiography. Source data are provided as a Source data file.

is rapidly cleaved, but Arg-ADPr is an ARH3 inhibitor with nanomolar affinity[28,52]. To understand the impact of the ARH3: substrate interaction on reaction efficiency, we crystallised ARH3 in complex with the chemically synthesised substrates: histone H2B peptide (aa 1–11) MARylated on Ser7 (H2BS7mar), dimeric ADP-ribose (dimer) and α-NAD$^+$ (Supplementary Tables 1, 2). Furthermore, we solved the structure of wt *Latimeria chalumnae* ARH3 (*Lch*ARH3) in complex with meADPr (Supplementary Tables 1, 2). Earlier described mutations inactivating ARH3 can be broadly classified into three categories affecting (i) protein stability, (ii) magnesium coordination or (iii) ADPr binding[15,23,27,28]. Some of these variants have been associated with stress-induced childhood-onset neurodegeneration with variable ataxia and seizures (CONDSIAS), an autosomal recessive disorder[22,23], and are annotated in the COSMIC database for cancer-associated mutations, which may indicate an influence on disease progression in some cancer types [https://cancer.sanger.ac.uk/cosmic][53] (Fig. 5e, Supplementary Fig. 4c and Table 3). Among the mutations we tested, three involved in adenosine coordination (F143L, S148A and G150E) and five within the active site (D34G, E41A, T76R, D77N and D78N) showed severely reduced catalytic activity (Fig. 5f and Supplementary Fig. 4b). We and others have

previously shown that mutation of Glu41 is compatible with the coordination of the Mg$_{II}$ ion as well as ligand binding[15,27,28]. Therefore, we chose to crystallise the hARH3 E41A substrate complexes as this prevents substrate cleavage during crystal formation, while retaining a functionally relevant magnesium coordination. The resulting structures show little divergence from the previous solved hARH3:ADPr complex (PDB 6D36) with r.m.s.d. of 0.284 Å (H2BS7mar), 0.316 Å (dimer), 0.176 Å (α-NAD$^+$) and 0.362 (meADPr) over 278, 270, 255 and 257 C$^\alpha$ atoms, respectively (Supplementary Fig. 6). All ligands are well defined in the electron density with the exception of the C-terminal peptide residues (Pro11 onwards) and ADPr$_{a1;n-1}$ distal phospho-ribose moiety of the dimeric PAR (Supplementary Fig. 7).

**ARH3 activation requires substrate-induced active site rearrangements.** Comparison of the active sites of our ARH3:substrate complexes revealed two striking features. First, substrate binding induces conformational flexibility within the Glu41-flap (composed of residues Glu41 to Tyr75), which was suggested to play a role in substrate entry and catalysis (Fig. 6a, Supplementary Fig. 8)[27,28]. In the closed conformation, Glu41 interacts with Mg$_{II}$, which induces tighter packing of the flap against the

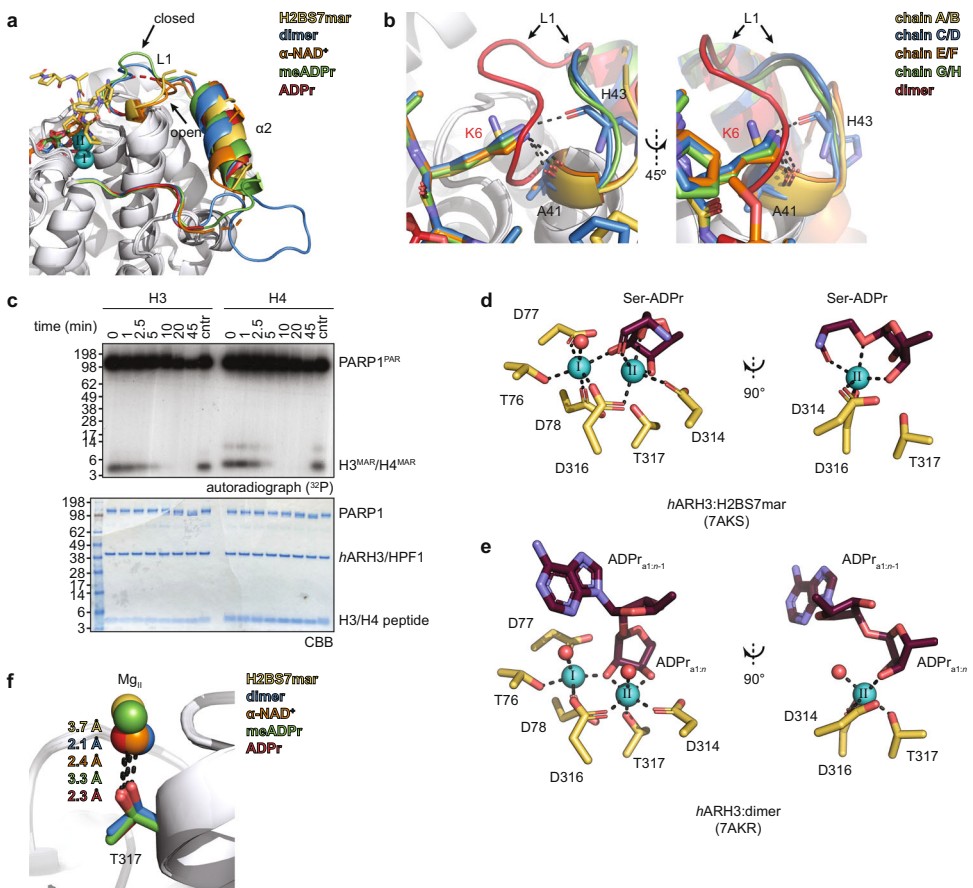

**Fig. 6 Distinct modes of substrate coordination by ARH3. a** Ligand-induced conformational changes in the Glu41-flap. Both H2BS7mar and α-NAD⁺ induce an open conformation characterised by the displacement of loop L1 and helix α2. In contrast, the Glu41-flap of the structures containing ADPr, PAR dimer and meADPr crystallised in a close conformation, thus allowing interaction of Glu41 with Mg_II (see also Supplementary Fig. 12). Ligand and Glu41-flap are given in corresponding colours and Mg²⁺ ions as turquoise spheres. Representation of the *h*ARH3:ADPr complex is based on PDB 6D36. **b** Peptide Lys6 can interact with the displaced loop L1 via polar backbone interaction (dashed lines) with Ala41 (Glu41 in wt *h*ARH3) and His43. For comparison closed Glu41-flap conformation of the *h*ARH3:dimer complex is given (red). **c** Presence of a KS motif is dispensable for ARH3 activity. Histone H3 and H4 peptides were enzymatically Ser-MARylated by PARP1/HPF1 in presence of ³²P-NAD⁺. Modification reversal by hARH3 was monitored as time-course experiment and representative results of two independent experiments are shown. **d**, **e** Influence of ligand binding on the coordination of the catalytic magnesium ions. Liquorice representation of ligand (purple) placement and magnesium (turquoise) coordination within the active site. Note, contacts and atoms involved in Mg_I coordination are omitted for clarity in the right panels. Water molecules are given as red spheres and coordination bonds are black dashed lines. **d** H2BS7mar provides three ligand atoms (3″O, 1″O [scissile bond] and S7 backbone-carbonyl oxygen) to the coordination of Mg_II, which leads to displacement of all water molecules from the Mg²⁺ ion as well as loss of the Thr317-Mg_II coordination (see (**f**)), thus adopting a strained square-pyramidal coordination. **e** Coordination of the PAR dimer is relaxed and both magnesium ions adopt an octahedral coordination. The scissile bond does not engage in the coordination and both Mg²⁺ ions coordinate an axial water molecule. **f** Comparison of ligand-dependent distances between Thr317 and Mg_II. The distance within the *h*ARH3:ADPr complex is based on PDB 6D36. Source data are provided as a Source data file.

domain core and was suggested to represent a post-catalytic state. In the open conformation, the Mg_II-Glu41 interaction is lost, which allows higher flexibility of the Glu41-flap, and was suggested to increase the accessibility of the active site[27]. Interestingly, while we observe an open conformation in the H2BS7mar- and α-NAD⁺-bound structures, both the *h*ARH3:dimer and *Lch*ARH3:meADPr complexes adopt a closed conformation (Fig. 6a). To exclude influences of crystal packing on Glu41-flap flexibility, we analysed the different crystal lattices (Supplementary Table 2) of our structures and found that in the majority of chains the Glu41-flap is orientated towards the intra-crystal solvent channels (Supplementary Figs. 8 and 9). Chain-to-chain comparison within the different structures showed that even in cases where Glu41-flap crystal contacts form, the overall arrangement is similar to chains without such contacts of the same enzyme state (Supplementary Fig. 8). Together, this suggests that crystal packing does not influence the observed Glu41-flap

conformations and the observed differences are mainly ligand-induced.

The AMP moiety of all four ligands is isostructurally orientated, whereas the distal riboses take up ligand-dependent poses (Supplementary Fig. 6g). The H2BS7mar peptide lies loosely coordinated within a groove running perpendicularly to the ADPr binding cleft (Supplementary Fig. 10a). It adopts no secondary structure and makes mostly water-mediated contacts with the *h*ARH3 protein. This binding mode is incompatible with the closed conformation of the Glu41-flap and indeed stabilises the open form via a Lys6 (side chain, H2B peptide):Glu41 (backbone, *h*ARH3) contact, which is part of the short serine-ADP-ribosylation consensus motif (KS) (Fig. 6b)[5,7]. To test whether the presence of a KS motif has an influence on the (ADP-ribosyl)hydrolase activity of ARH3, we performed ADP-ribosylation hydrolysis experiments using previously established peptide substrates: H3 (aa 1–21), which contains an internal

modification site (Ser10) within the consensus KS motif, and H4 (aa 1–23), which is primarily modified on the N-terminal Ser1[20]. Under the assay conditions both peptides show comparable modification efficiency and similar de-modification progression (Fig. 6c). While we cannot fully exclude contribution to substrate selection of the KS lysine or other nearby residues, these results suggest that the KS lysine is dispensable for efficient ADP-ribosylation hydrolysis and that ARH3 has few limitations regarding the sequential context of the modification. This is in line with the observed broad target spectrum of ARH3 following DNA damage[29]. In addition, the broad target spectrum suggests that substrate binding depends primarily on the ADPr moiety attached to the protein substrate. Concordantly, mutations of Phe143 and Ser148 (F143L and S148A), which coordinate the adenine base, show almost complete loss of catalytic activity. In contrast, the structurally adjacent COSMIC mutations S185P and L186V appear fully active (Fig. 5f). Mutation of residues along the peptide channel (Gly115, Asn269, Gly270, and Ile271) do not alter the enzymatic activity (Fig. 5f). Note, mutation of Tyr149 to leucine, the second residue π-stacking with the adenine base, does not influence enzymatic activity, most likely due to retention of tight packing of the adenosine in this mutant.

Our structural data further reveal that in the α-NAD+ complex Glu41 would be displaced from the axial $Mg_{II}$ position by the nicotinamide moiety, thus forcing the observed open conformation (Supplementary Fig. 12d). In contrast, the $ADPr_{a1:n}$ unit of the dimer binds in a relaxed conformation within the ligand site, which prevents engagement of the scissile bond with the $Mg^{2+}$ ions (Fig. 6e, Supplementary Figs. 10b, 12c). Similar to the peptide substrate, the $ADPr_{a1:n-1}$ makes only water-mediated contacts with the protein. While the adenosine base appears well defined with only slightly elevated temperature values (B-factors), the latter increase sharply towards the phosphate groups (Supplementary Fig. 11). The different conformations adopted by the phosphates together with the lack of reliable electron density for the distal ribose further indicate a high degree of flexibility within this part of the molecule. The axial position of $Mg_{II}$ is occupied by a water molecule (w610) due to the E41A mutation that was introduced to allow crystallisation. However, the protein backbone adopts a closed conformation, and our models suggest an interaction between Glu41 and $Mg_{II}$ is possible (Supplementary Fig. 10b). This is also in accordance with the observed interaction of Glu33, the homologous residue to Glu41 in *L. chalumnae*, and $Mg_{II}$ in the *Lch*ARH3:meADPr complex (Supplementary Fig. 10c).

Second, complexes of H2BS7mar, α-NAD+ and meADPr show both altered coordination and water displacement from $Mg_{II}$. H2BS7mar and meADPr act similarly to chelants via the 1″ and 2″ oxygens (Fig. 6d, e and Supplementary Fig. 12b, e). In addition, the backbone-carbonyl of peptide Ser7 occupies the axial position of $Mg_{II}$. In the apo form (PDB 2FOZ) and the *Lch*ARH3:meADPr complex this position is occupied by Glu41 (Glu33 in *Lch*ARH3), which stabilises a closed conformation (Supplementary Fig. 12a, d). The interactions with either H2BS7mar or meADPr increase the Thr317-$Mg_{II}$ distance by 0.9–1.6 Å relative to the ADPr-bound state, which breaks the coordinating bond and leads to an unusual five-coordinated distorted square-pyramidal geometry around the $Mg^{2+}$ ion (Fig. 6d, f and Supplementary Fig. 12b, e). Within the *h*ARH3:α-NAD+ complex, the interaction between nicotinamide and $Mg_{II}$ is sterically dominated by the binding of the ADPr moiety within the active site. The positive charge at the N1 nitrogen and long $Mg_{II}$-ring atom distances (3.4–4.1 Å; Supplementary Fig. 12e and Supplementary Table 5) suggest that the arene cannot act like a chelant to displace $Mg_{II}$ from the protein coordination. However, the nicotinamide moiety sterically prevents occupation of the axial ligand position of $Mg_{II}$, thus

creating a free valence at the $Mg^{2+}$ ion and forcing the Glu41-flap to open. In addition, the nicotinamide positioning is distinctly asymmetric, reminiscent of ring slippage, which may polarise the π electron system and weaken the N-glycosidic bond (Supplementary Table 5). Noteworthy, neither the H2BS7mar nor the α-NAD+ complex have a water ligand at $Mg_{II}$, thus excluding the ion as a water activator during the reaction cycle (Fig. 6d and Supplementary Fig. 11d).

**Glu41 positioning leads to substrate activation**. Given that the E41A mutation allows transition between the open and closed conformation while remaining catalytically inactive, the maintenance of a closed conformation in the absence of substrate cannot be the only function of this residue. Our structural analysis suggests that this residue is required for further substrate activation following binding. In the *h*ARH3:H2BS7mar complex, Glu41 is shielded from the aqueous environment by the peptide and is no longer able to form a first-order coordination bond with $Mg_{II}$. In the structure, we observed a water molecule (w592; chain A) residing within the second coordination sphere of $Mg_{I}$, which coincided with the carboxylate group of a naïvely placed favourable glutamate rotamer. Energy minimisation modelling of this conformation into the water density leads to the formation of second coordination sphere interactions with both $Mg^{2+}$ ions (Fig. 7a and Supplementary Fig. 13). This conformation would allow interaction with the axial water ligand of $Mg_{I}$, which in turn could activate and position the molecule for a nucleophilic attack from the scissile bond oxygen. While a similar Glu41 conformation in the α-NAD+ complex is possible, the differences between the O- and N-glycosidic bonds render an initial nucleophilic attack on α-NAD+ superfluous for hydrolysis. Furthermore, the absence of the peptide backbone imposes less restrictions on potential Glu41 side-chain conformations and an interaction between Glu41 and the pyridine ring amide appears feasible. This interaction could fix the relative positioning of the ring towards the magnesium ion and further withdraw electrons from the ring system, thus contributing to the weakening of the nicotinamide-ribose bond.

Taken together our data support a mechanism in which both the achievement of a high energy $Mg_{II}$ state as well as a repositioning of Glu41 are prerequisites to catalysis. The former is demonstrated by the inability of *h*ARH3 to cleave meADPr despite the presence of an only five-coordinated magnesium ion (Supplementary Figs. 4a and 12d). This lack of activity against meADPr, however, raises the question why OAADPr can be efficiently cleaved by ARH3[50,54]? While we were unable to co-crystallise ARH3 with 1″-O-AADPr, which is the confirmed substrate isomer, due to transesterification, which disfavours this linkage under our crystallisation conditions, our data strongly suggest that this ligand would need to displace Glu41. Most likely displacement is achieved by a $Mg^{2+}$:substrate interaction that does not involve the scissile bond oxygen, but rather the adjacent carbonyl group. We modelled this interaction and found that the scissile bond would be orientated towards the water ligand of $Mg_{I}$, thus favouring a nucleophilic attack (Fig. 7b and Supplementary Figs. 12g and 14). In addition, this interaction would place the terminal carbon of the acetate facing away from the peptide binding site, disfavouring the binding of peptides containing Asp- or Glu-linkages within the active site and thus providing an explanation for the inability of ARH3 to cleave these modifications[12].

**ARH3 reaction progresses through an oxocarbenium ion transition state**. To clarify reaction progression post initiation, we aim to distinguish between the proposed formation of an

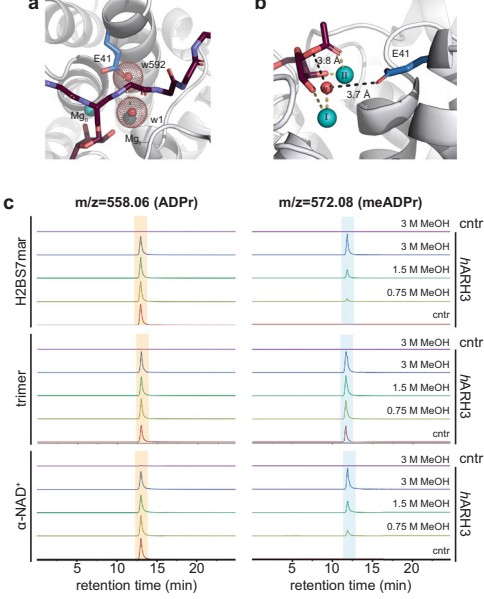

**Fig. 7 Analysis of reaction initiation and progression in the ARH3 catalytic cycle. a** Model of *h*ARH3 wt in complex with H2BS7mar. Glu41 (blue) adopts a conformation within the second coordination sphere of both $Mg^{2+}$ ions. Water molecules shown (red sphere) belong to the model (w1, axial water) or to the template *h*ARH3:H2BS7mar structure (w592, chain A), respectively. Side-chain atoms of the peptide (purple) were omitted for clarity. **b** Model of *h*ARH3 wt in complex with *O*AADPr. The axial water of $Mg_I$ (w1; red sphere) was included into the model. Selected magnesium coordination bonds are shown in yellow and distances between model Glu41, w1 and scissile bond oxygen are shown in black. **c** High-resolution mass spectrometry analysis of solvolysis of H2BS7mar, trimeric PAR and α-NAD⁺ in the presence and absence of methanol. Shown are the total negative ion counts for ADPr $[M-H]^-$ with a *m/z* of 558.06 (left panels) meADPr $[M-H]^-$ with a *m/z* of 558.06 (right panels). Ion chromatograms were extracted setting tolerance at 10 ppm. Note, that trimeric PAR contains meADPr, which is released upon cleavage and increases experimental background. Source data are provided as a Source data file.

oxocarbenium ion or seryl alkoxide[10,27–29] intermediate by performing solvolytic experiments in presence of methanol as alternative nucleophile. Methanol has a higher propensity to react with oxocarbenium ions than water and meADPr should be observable as reaction product if the reaction progress through this intermediate[55]. Indeed, we observed the ARH3-dependent formation of meADPr regardless of the used substrate (H2BS7mar, PAR trimer and α-NAD⁺; Supplementary Fig. 7c). This commonality in the intermediate formation suggests that, while difference in reaction initiation may exist, the later steps of the ARH3 catalytic cycle are substrate independent. Future work is needed to determine whether the reaction progresses step-wise with a solvent-equilibrated intermediate or via a preassociated/ concerted mechanism with an oxocarbenium ion-like transition state[56].

## Discussion

The dynamics of PAR signal turnover are a crucial determinant of its physiological outcomes. However, the complexity of the PAR molecule with the apparent stochastic nature of branch point introduction has long hampered efforts to study its degradation directly. Thus far, only linear PAR fragments were available either by liquid chromatographic purification of enzyme-derived

polymers or chemical synthesis[32,57]. Due to the unavailability of defined branch points, observations of their cleavage have relied on the analysis of mixed, enzyme-derived polymers to infer hydrolase properties[26,58]. Here we describe the de novo synthesis of branched point polymers, thus making this elusive PAR building block available for targeted studies. We utilised this approach to confirm that the PAR degradation step is primarily driven by PARG, which can efficiently cleave linear and branched polymers. Long incubations with a high ARH3 concentration showed detectable activity against branch points, but this very low activity may be physiologically insignificant. Our findings suggest that introduction of PAR branch points imposes a further level of specific regulation: not only providing these structural binding platforms for specialised binders, such as APLF[31], but also regulating signal duration, thus altering physiological outcomes. This is in line with recent studies that show a correlation between branch frequency and physiological response[30,31].

While PARG is only able to cleave the *O*-glycosidic PAR bond, ARH3 is known to process chemically distinct substrates[12,16,51,54,59]. Amongst the more surprising findings in substrate selection and turnover is that ARH3, despite the similarities in bond nature, is able to process Ser-ADPr and *O*AADPr efficiently, whereas PAR degradation by ARH3 is a much slower process[12,16]. Our structural work gives rise to mechanistic insights into these differences and the molecular factors governing substrate activation and hydrolysis initiation.

First, the substrates directly interact with the magnesium centre and induce a square pyramidal coordination geometry at $Mg_{II}$. This unusual coordination can be found in several chemical compounds but has so far only been observed twice in protein structures[60,61]. The NG GTPase domain of the Ffh subunit of the prokaryotic signal recognition particle (SRP) contains a mononuclear, catalytic magnesium centre. In the GDP-bound form, the ion is usually octahedrally coordinated by four water molecules, the GDP β-phosphate and a threonine side-chain oxygen. However, a second, square planar, state was observed, which was stabilised through rotation of a glutamate carboxyl into the second coordination sphere (PDB 1O87). These changes in the metal coordination share similarities with our observed H2BS7mar structure; however, in the Ffh GTPase a post-catalytic state was represented, as the GTPase substrate activation involves octahedral coordination of β- and γ-phosphate at the magnesium centre. Our structural models predict that Glu41 enters the second coordination sphere of both $Mg^{2+}$ ions after displacement from its axial $Mg_{II}$ coordination position, which in wt ARH3 may help stabilise the observed five-coordinated intermediate. Furthermore, Glu41 positioned in this way may activate the axial water at $Mg_I$ for a nucleophilic attack onto the 1″O (Figs. 4a, 5). The second example of a square pyramidal coordination was observed in a Lon AAA+ protease (LonA) crystallised in presence of magnesium and the covalent inhibitor bortezomib (PDB 4YPM)[61]. Attachment of the inhibitor occurs in close proximity to the metal-binding site and sterically blocks access of water molecules to one of the ligand sites. This mirrors the binding configuration we observe in our α-NAD⁺ structure in which the nicotinamide ring sterically prevents access of water or Glu41 to the $Mg_{II}$ centre. The relative distance between magnesium ion and nicotinamide in our structure suggests that rather than engaging in metal coordination, the ligand displacement leads to a free valence and a high energy magnesium state. Together with slippage-like positioning of the nicotinamide ring, which introduced polarisation into the π electron system, it likely provides the energy for bond cleavage.

Second, the substrate needs to induce the open conformation of ARH3 through displacement of Glu41 from the $Mg_{II}$ coordination. Such a displacement not only allows the engagement of

the substrate with the magnesium centre but also leads to the repositioning of Glu41 into the catalytic position. The importance of this step can be observed in our *Lch*ARH:meADPr structure in which the ligand engages with the magnesium centre, but fails to displace Glu33 (Glu41 in *h*ARH3) from $Mg_{II}$. Therefore, meADPr can act as an inhibitor of ARH3 instead of being hydrolysed. Our structures furthermore suggest that this engagement is the rate-limiting step in the hydrolysis of PAR. Within our *h*ARH3:dimer structure, crystal packing does not impact the positioning of the $ADPr_{a1:n-1}$ moiety. Both ADPr groups of the dimeric PAR adopt a relaxed conformation and do not engage with the $Mg_{II}$ centre to displace Glu41. The most likely reason for this unproductive binding mode is the strained position the $ADPr_{a1:n-1}$ would need to adopt as well as the lack of a further coordination groups (like the serine backbone carbonyl) to stabilise the binding.

It is worth noting that earlier apo and ADPr-bound structures of *h*ARH3 showed a μ-aqua ligand at the metal centre (Supplementary Fig. 12a)[27,29,62], which was suggested to play a catalytic role[29]. We were, however, unable to observe this ligand in any of the here reported structures (Supplementary Fig. 12) or in our earlier reported structures of *Lch*ARH3 product (analogues) complexes[28]. This absence, together with the unusually short coordination bonds observed in the earlier structures, suggests that the μ-aqua ligand is dispensable for catalysis and may have rather a function in the stabilisation of the binuclear magnesium centre in absence of a substrate or ligand.

Together these observations clarify the initial steps of the reaction (Fig. 8), but the reaction progression post glycosidic bond cleavage is not immediately obvious and depends on which of a variety of possible intermediates, including the earlier proposed oxocarbenium ion and seryl alkoxide[27–29], is formed.

Introduction of methanol to the reaction leads to the formation of meADPr, thus strongly suggesting that an oxocarbenium intermediate or a transition state with strong oxocarbenium ion-like characteristic is formed. It is a generally held view that oxocarbenium ions are too unstable to exist as a free intermediate and hence stabilisation in the reaction cycle would be expected[56]. However, we could not observe any suitably coordinated nucleophilic water, needed for the resolution of a free oxocarbenium ion via an attack on the $C1''$ position, or stabilising residues in our crystal structure. The steric limitation within the active site further suggests that the nucleophilic attack can only be realised from the α-face of the ribose, which, unlike in macrodomain-type hydrolysis, preserves the configuration at the stereocentre. A likely explanation could be that the required water molecule is part of the outer magnesium coordination spheres and thus prone to enter the active site following coordination changes during substrate turnover (Fig. 8).

In conclusion, our synthesis strategy for branch points within PAR polymers made it possible to gain insights into the substrate selection and hydrolysis of PARG and ARH3. The synthesis strategy is scalable and can be used in future studies to overcome limitations in the availability of well-defined substrates that has long hampered investigations in this important area. Here we provide a more complete understanding of PAR removal from branch point cleavage to serine MARylation hydrolysis. The reduced activity of PARG and ARH3 against branch points observed in this study provides a plausible explanation for the phenotypically observed impact of branch frequency on cellular outcomes of ADP-ribosylation signalling[30,31]. Furthermore, our findings give further insights into the molecular mechanism of abnormal ADP-ribosylation signalling identified in several

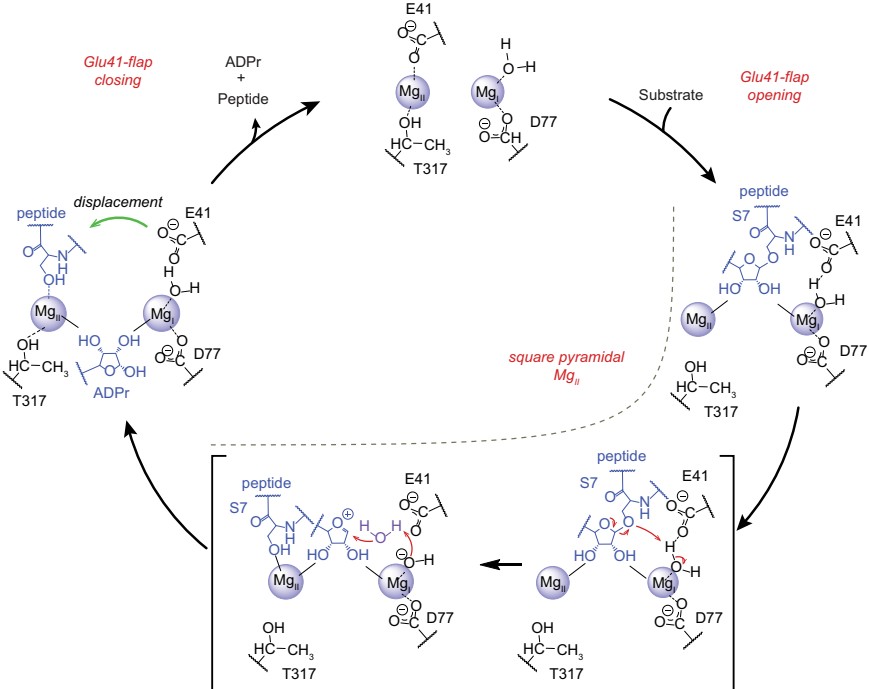

**Fig. 8 Proposed reaction mechanism for ARH3.** In the resting state, ARH3 adopts a closed conformation, which allows interaction of Glu41 (human) with $Mg_{II}$. Binding of the ligand occurs preliminary via the ADP-ribose moiety and the ligand needs to displace Glu41 and engage with $Mg_{II}$ to form the pre-catalytic complex. This triggers a conformational change into the open conformation and alters the coordination geometry of $Mg_{II}$ from octahedral to square pyramidal. The displaced Glu41 moves into the second coordination sphere of both magnesium ions and assists in activation of axial water coordinated at $Mg_I$ for a nucleophilic attack on $1''C$. Cleavage of the glycosidic bond leads to the formation of a oxocarbenium intermediate, which is resolved by a solvent-derived nucleophilic water (purple). Depicted is a step-wise solvent-equilibrated intermediate, but a preassociated/concerted transition state is conceivable. Furthermore, cleavage of the glycosidic bond allows for $Mg_{II}$ to transition back to an octahedral coordination and the de-modified serine product can be displaced from $Mg_{II}$ by Glu41, which results in closure of the active site.

diseases. For instance, (ADP-ribosyl)hydrolase deficiencies readily lead to neurodegeneration[22,23,63,64] and in particular ARH3 was associated with alteration of the histone code[20] and failure to revert back to a pre-damage chromatin state after DNA repair[21].

## Methods

**Plasmid construction.** Expression vectors for *hARH3*, *LchARH3*, HPF1, PARP1 and PARG were described earlier[3–7]. All indicated mutations were introduced via PCR based site-directed mutagenesis (Supplementary Table 6). The *hARH3* E41A crystallisation construct was generated by PCR based site-directed mutagenesis altering the encoded N-terminal sequence: parts of the linker region as well as aa 1–18 of ARH3 were replaced by a HRV3C cleavage site (N-terminal protein sequence of construct utilised in the biochemical assays MGSHHHHHHDITS LYKKAGSAAAVLEENLYFQGSFTMAAAAMAAAAGGGAGAARSLSR[…] was altered to MGSHHHHHHDITSLEVLFQGPGSSLSR[…] for crystallisation) (Supplementary Table 6).

### Protein expression and purification

*For biochemistry.* Expression of recombinant proteins in Rosetta (DE3) cells grown in LB medium was induced at $OD_{600}$ 0.6 with 0.4 mM IPTG, cells were grown overnight at 290 K and harvested by centrifugation. Recombinant His-tagged proteins were purified at 277 K by $Ni^{2+}$-NTA chromatography (Jena Bioscience) according to the manufacturer's protocol using the following buffers: all buffers contained 50 mM TrisHCl (pH 8) and 500 mM NaCl; additionally, the lysis buffer contained 25 mM, the washing buffer 40 mM, and the elution buffer 500 mM imidazole. For purification of the ARH3 proteins all buffers also contained 10 mM $MgCl_2$. All proteins were dialysed overnight against 50 mM TrisHCl (pH 8), 200 mM NaCl, 1 mM DTT and 5% (v/v) glycerol.

PARP1 and its mutants were transformed in Rosetta (DE3) cells grown in 2xYT medium supplemented with 10 mM benzamide and expression was induced at OD600 0.6 with 0.4 mM IPTG, cells were grown overnight at 290 K and harvested by centrifugation. Pellets were resuspended in lysis buffer (25 mM HEPES [pH 8], 500 mM NaCl, 0.5 mM TCEP) and lysed by high-pressure homogenisation. The proteins were purified using on an ÄKTA FPLC system (GE Healthcare Life Sciences) at 277 K via affinity chromatography using a HisTrap HP column (GE Healthcare Life Sciences). After lysate application, the column was washed with lysis buffer supplemented with, first, 50 mM imidazole and, second, NaCl to a total concentration of 1000 mM, followed by elution of bound protein with the addition of 250 mM imidazole. The eluate was diluted fivefold with 25 mM Tris, pH 7, 100 mM NaCl, 0.5 mM TCEP, applied to a HiTrap Heparin column (GE Healthcare Life Sciences) equilibrated in the same buffer, eluted with a linear NaCl gradient to 1000 mM. Fractions containing PARP1 were pooled and loaded on a Superdex 200 Increase 10/300 GL, and eluted with 200 mM NaCl, 0.5 mM TCEP.

*For crystallisation.* LchARH3 wt and hARH3 E41A underwent affinity purification over a HisTrap HP column (GE Healthcare), followed by dialysis against 50 mM TrisHCl (pH 8), 500 mM NaCl, 20 mM imidazole and 1 mM DTT in presence of HRV3C protease for proteolytic cleavage of the His-tag. Removal of the uncleaved protein was achieved by rebinding to a HisTrap HP column and the protease was removed by binding to a GSTrap 4B column (GE Healthcare). The final step involved size exclusion chromatography using a HiLoad Superdex 75 pg column with 10 mM TrisHCl (pH 8), 100 mM NaCl and 1 mM DTT (hARH3) or 10 mM PIPES (pH 7), 100 mM NaCl and 1 mM DTT (LchARH3) as elution buffer.

### Chemical synthesis

The synthesis of linear ADPr polymers and the H2BS7mar peptide were described earlier[32,36]. The synthesis of poly(ADP-ribosyl) branch points and meADPr as well as their validation is described in the Supplementary Methods.

### (ADP-ribosyl)hydrolase activity assays

*Enzymatic activity assay using thin-layer chromatography (TLC).* 2 mM substrate, either synthetic branch point or linear trimer, was incubated with 10 µM ARH3 or PARG as indicated in assay buffer (50 mM TrisHCl [pH 7.5], 200 mM NaCl, 1 mM $MgCl_2$, 1 mM DTT) at 30 °C for 45 min. Reactions were transferred on ice and 1 µL spotted onto Silica gel containing 254 nm fluorescence indicator on aluminium support (Fluka Analytica, Sigma-Aldrich), dried and developed in 0.2 M $NH_4HCO_3$ in ethanol:water 7:3 (vol/vol). Reaction products were visualised under UV light. For time-course experiments reactions were stopped either by addition of 25 mM EDTA (pH 8; ARH3) or 100 µM PDD00017273 (PARG; Tocris).

*Enzymatic activity analysis using luminescence detection of ADPr.* ARH3 mutant analysis was performed essentially as described in ref. 36. Briefly, 20 µM substrate (branch point, trimer or H2BS7mar peptide) was hydrolysed in presence of 1 µM ARH3 or 0.5 µM PARG (wt or mutant as indicated) in assay buffer (50 mM TrisHCl [pH 7.5], 200 mM NaCl, 10 mM $MgCl_2$, 1 mM DTT and 0.23 µM NUDT5) for 45 min at 30 °C. Reactions were analysed using AMP-Glo™ assay kit (Promega) according to manufactures recommendations and luminescence

recorded using a SpectraMax M5 plate reader with the SoftMax Pro software (Molecular Devices). Data were analysed using Prism (v9.1, GraphPad).

*Enzymatic activity assay using PARP1-derived MAR and PAR.* ARH3 activity assays were performed essentially as described in[12]. Briefly, histone H3 (aa 1–20, biotinylated) or H4 (aa 1–23, biotinylated) peptides were modified by incubation with 0.5 µM PARP1, 0.6 µM HPF1 and activated DNA (Trevigen) in assay buffer (50 mM TrisHCl [pH 8], 200 mM NaCl, 1 mM $MgCl_2$, 1 mM DTT, 25 µM $NAD^+$ and 1 µCi $^{32}P$-$NAD^+$). Reactions were incubated for 30 min at 30 °C and stopped by addition of 100 µM olaparib. To generate modified PARP1 mutants, the same reactions were performed excluding the histone peptides and utilising a higher $NAD^+$ concentration (50 µM). Reactions were further incubated in presence of 1 µM hydrolase for 45 min at 30 °C. Reactions were stopped by addition of LDS sample buffer (Life Technologies) and incubation at 95 °C for 3 min. Samples were then analysed by SDS-PAGE and autoradiography. Alternatively, reaction were performed as described in absence of $^{32}P$-$NAD^+$ and analysed by SDS-PAGE and immunoblot using rabbit monoclonal anti-pan-ADP-ribose binding reagent (Millipore, MABE1016; RRID: AB_2665466) and mouse monoclonal anti-6xHis antibody (Takara, 631212; RRID: AB_2721905) as primary antibodies and polyclonal goat anti-mouse immunoglobulins/HRP (Dako, P0447; RRID: AB_2617137) and polyclonal swine anti-rabbit immunoglobulins/HRP (Dako, P0399; RRID: AB_2617141) as secondary antibodies. For inhibitor study, the ARH3 was preincubated with the indicated amount of inhibitor for 5 min at RT.

### Liquid chromatography-high resolution mass spectrometry (LC-HRMS)

*ARH3 and PARG catalysed hydrolysis.* Samples were prepared and analysed by TLC (see above) and 6 µL of the reaction were diluted 1:10 in $dH_2O$. Proteins were removed by filtration using Amicon Ultra centrifuge filters (3000 MWCO; Sigma-Aldrich). Flow-through was stored at −80 °C until analysis.

*ARH3 catalysed solvolysis.* 300 µM substrate (H2BS7mar peptide, synthetic trimer or α-$NAD^+$) was incubated with 4 µM ARH3 in assay buffer (50 mM TrisHCl [pH 7.5], 200 mM NaCl, 1 mM $MgCl_2$, 1 mM DTT) and presence of 0.75, 1.5 or 3 M methanol (as indicated) at 30 °C for 2 h (H2BS7mar and α-$NAD^+$) or 12 h (trimer). The reaction was stopped by protein removal through filtration using Nanosep® centrifuge filters (Omega membrane, 3000 MWCO; Pall Corporation). Samples were subsequently stored at −80 °C until analysis.

*LC-HRMS analysis.* ADPr and other nucleotides were analysed by a method previously developed in ref. 65. Briefly, a Thermo Exactive mass spectrometer equipped with Waters Acquity liquid chromatography system was used for high-resolution mass spectrometry. Thermo Xcalibur software was used for controlling the instrument. Mass resolution was set to ultra-high (100,000 at 1 Hz) according to the manufacturer manual to achieve measurement accuracy of 0.002 FWHM at *m/z* 200.0000. Before each measurement, the system was calibrated and its accuracy with external calibration was confirmed to be better than 5 ppm. We adjusted the Electrospray source conditions to maximise sensitivity, and detection mode was set to detect both positive (+) and negative (−) ions. UV-visible chromatograms were recorded at 260 nm. Five microlitre of sample volumes were separated over a SeQuant® ZIC®-HILIC column (100 × 2.1 mm, 5 µm particle size, 200 Å pore size; Merck) using the following buffer systems: buffer A contained 20 mM ammonium acetate (pH 7.5) in MeCN (Honeywell, CHROMASOLV® 99.9%):water 9:1 (vol/vol) and buffer B 20 mM ammonium acetate (pH 7.5) in water. The column was equilibrated to 40 °C and sample eluted at 0.2 mL/min using the method: [0–1 min]: 100:0 (A:B); [1–22 min] 100:0 (A:B) linearly changed to 50:50 (A:B); [22–25 min] 50:50 (A:B); [25–26 min] 50:50 (A:B) linearly changed to 80:20 (A:B); [26–30 min] 80:20 (A:B); [30–31 min] 80:20 (A:B) linearly changed to 100:0 (A:B); and [31–45 min] 100:0 (A:B). All compounds were eluted within 25 min and the gradient was kept 100:0 (A:B) from 31–45 min to assure the column return to equilibration before the next injection. MS data were recorded from 0–25 min. Analysis of the LC-HRMS data was performed using MNova software (v13, Mestrelab Research).

### Analysis of branch frequency of in vivo generated PAR

UPLC-MS/MS analyses of PAR were performed as described previously[44,45,66], with some modifications. Control and ARH3$^{−/−}$ U2OS cells were cultured in DMEM (Sigma) containing 10% FBS and penicillin/streptomycin (both Thermo Fisher Scientific) and treated with DMSO or 25 µM PARGi for 4 days. Afterwards, the media was removed, cells were washed with PBS and lysed by addition of ice-cold 20% TCA and detached mechanically using cell scrapers. Precipitates were pelleted by centrifugation at $3000 \times g$ for 5 min, pellets washed with 70% ice-cold EtOH, air-dried for about 1 h at 37 °C, and stored at −20 °C until further processing. To detach protein-bound PAR, samples were dissolved in 255 µl 0.5 M KOH for ~1 h at 37 °C and 400 rpm on a thermomixer (Eppendorf), and subsequently neutralized by addition of 50 µl 4.8 M MOPS buffer. A 30 µl aliquot of the solution was stored at −80 °C for DNA concentration determination. To the rest of the sample, 10 µl of 1.2 µM $C^{13}$, $N^{15}$-labelled PAR standard was added, and DNA and RNA were digested by adding 6.25 µl 2 M $MgCl_2$, 2.5 µl 100 mM $CaCl_2$, 12.5 µl 2 mg/ml DNase (Roche) and 2.5 µl 1 mg/ml RNase (Thermo Fisher Scientific), and incubated for 3 h at 37 °C and

300 rpm in a thermomixer. Afterwards, 1.25 μl of 40 mg/ml proteinase K (Roche) was added and samples were incubated at 37 °C and 300 rpm overnight. Then, PAR was purified using the High Pure microRNA Isolation kit (Roche) according to the manufacturer's instructions. Briefly, 300 μl of sample were mixed with 624 μl Binding Buffer and 400 μl Enhancer and loaded onto the column assembly (High Pure Filters on Collection Tubes). The column assembly was centrifuged at 15,700 × g for 30 s, washed once with 300 μl and once with 200 μl Wash Buffer and finally centrifuged again to dry columns completely. PAR was eluted by adding 100 μl Milli-Q $H_2O$ and centrifuging again for 1 min. To digest purified PAR to nucleosides, samples were incubated for 3 h at 37 °C and 400 rpm on a thermomixer in a solution containing 10 U alkaline phosphatase (Sigma), 0.5 U phosphodiesterase (Affymetrix), 1.4 mM $Mg(Ac)_2$ and 34 mM $NH_4Ac$. Afterwards, samples were loaded onto Nanosep 10 K Omega columns (Pall) and centrifuged at 15,700 × g for 10 min. Samples were dried in SpeedVac vacuum concentrator Univapo 100 ECH (Uni Equip) and resuspended in 25 μl Mill-Q $H_2O$ prior to MS measurement. After centrifugation at 15,700 × g for 5 min, 20 μl of each sample were transferred to an MS vial. Subsequently, ribosyl-adenosine (R-Ado) and diribosyl-adenosine (R₂-Ado) were quantified by isotope dilution UPLC-MS/MS. Therefore, 15 μl of the sample was injected into an ACQUITY UPLC H-Class which was coupled to a Xevo TQ-S triple quadrupole mass spectrometer (Waters). The sample components were separated via an ACQUITY UPLC BEH C18 column, 130 Å, 1.7 μm and 2.1 mm × 50 mm (Waters) with an isocratic gradient of 99% solvent A (water with 0.01% formic acid) and 1% solvent B (acetonitrile with 0.01 % formic acid) and a flow of 0.5 ml/min over for 10 min. The column temperature was held at 30 °C. Molecules were ionized using electron spray ionization in the positive ion mode. R-Ado ($m/z$ 400 → 136), $C^{13}$, $N^{15}$-labelled R-Ado ($m/z$ 415 → 146), R₂-Ado ($m/z$ 532 → 136), and $C^{13}$, $N^{15}$-labelled R₂-Ado ($m/z$ 547 → 146) were analysed in the mass spectrometer via multiple reaction monitoring using instruments settings as shown in Supplementary Table 7.

Data were analysed using Prism (v9.1, GraphPad).

**Crystallisation**. For crystallisation, *h*ARH3 was expressed as described above and purified protein was concentrated to 270 μM (~10.2 mg/mL) in the final crystallisation buffer containing 0.9 mM $MgCl_2$ and 1.8 $CaCl_2$ and either 1.5 mM H2BS7mar, 1.5 mM PAR dimer or 2.7 mM α-NAD⁺. Crystals containing PAR dimer and α-NAD⁺ were identified by the sitting-drop vapour diffusion method in MRC 96 well plates (Molecular Dimensions) at 292 K using the LMB Crystallisation and Stura FootPrint Combination HT-96 screens (Molecular Dimensions), respectively. Final crystallisation conditions contain (a) *h*ARH3:H2BS7mar crystals were grown in sodium citrate (pH 6.1), 18% (w/v) PEG4000 and 400 mM ammonium acetate; (b) *h*ARH3:PAR dimer crystals were grown in 100 mM TrisHCl (pH 8.5), 20% (w/v) PEG4000 and 200 mM $MgCl_2$; (c) *h*ARH3:α-NAD⁺ crystals were grown in 100 mM ammonium acetate (pH 4.5) and 9% (w/v) PEG10000.

*Lch*ARH3 for crystallisation was expressed as described above and purified protein was concentrated to 300 μM (~11.5 mg/mL). Crystals for structure determination and soaking experiments were grown at 292 K by the sitting-drop vapour diffusion method in MRC 96 well plates (Molecular Dimensions) in 100 mM sodium citrate (pH 5), 22% (w/v) PEG4000 and 200 mM ammonium acetate. Apo crystals were soaked with 3 mM meADPr for 45 min in mother liquor containing 10 mM $MgCl_2$ and 10% (v/v) ethylene glycol.

All crystals were vitrified by transfer into mother liquor supplemented with 16% (v/v) ethylene glycol for 5 s prior to submersion in liquid nitrogen.

**X-ray data collection, processing and refinement**. X-ray diffraction data were collected with the in-house Generic Data Acquisition (GDA) software using synchrotron radiation at Diamond Light Source (Rutherford Appleton Laboratory, Harwell, UK) (Supplementary Table 1). Datasets of *h*ARH3 in complex with H2BS7mar, dimer and α-NAD⁺ were collected cryo-cooled at 100 K using 0.9686 Å wavelength at beamline I24 and a dataset of the *Lch*ARH3:meADPr complex were collected cryo-cooled at 100 K using 0.9762 Å wavelength at beamline I03. Data were processed using Xia2[67]. *h*ARH3 phases were solved by molecular replacement using PHASER[68] as implemented in the CCP4i2 package[69] using as search model human ARH3 (PDB 6D36) and *Lch*ARH3 (PDB 6HH3). The molecular replacement solutions were refined by iterative cycles of manual structure building using Phenix[70], REFMAC5[71] and Coot[72]. Structures were validated using MolProbity[73] and figures were prepared using PyMOL (Molecular Graphics System, Version 2.3 Schrödinger, LLC). Ramachandran statistics ([%] favoured/allowed/disallowed): *h*ARH3: H3BS7mar (98.19/1.81/0.0), *h*ARH3:dimer (98.07/1.93/0.0), *h*ARH3:α-NAD⁺ (98.15/1.85/0.0) and *Lch*ARH3:meADPr (97.69/2.31/0.0).

**Structural modelling of PARG and ARH3**

*PARG*. We used PARG:ADPr dimer (PDB 5A7R) as starting point to generate PARG:branch point models (Supplementary Fig. 3). The branched ADPr molecule was designed and built using Ligand Builder integrated in Coot[72] and fitted in the ADPr dimer density of A5R7, so that either ADPr_{a2:1} (model 1) or [ADPr_{b1:1}] (model 2) lay iso-structural to ADPr_{a1:n} (PDB 5A7R) in the ligand-binding site. Energy minimisation of these models was performed in Yasara[74], keeping the

ADPr moiety placed in the catalytic pocket fixed and allowing the rest of the molecule to minimise.

*ARH3*. We used *Lch*ARH3:meADPr (PDB 7AQM) and *h*ARH3:H2BS7mar (PDB 7AKS) as a template for modelling of the OAADPr ligand. OAADPr was designed and built using Ligand Builder integrated in Coot[72] and fitted in the meADPr density. For the reverse mutation A41E, acceptable rotamer conformation of the glutamate residue was generated in Coot. Subsequently, all atoms of both models were energy minimised in Chimera (http://www.rbvi.ucsf.edu/chimera)[75]. The parameters used were: 10 steps of steepest descent minimisation (0.02 Å step size), 10 steps of conjugate gradient (0.02 Å step size), 10 of update interval.

**Reporting summary**. Further information on research design is available in the Nature Research Reporting Summary linked to this article.

## Data availability

The atomic coordinates and structure factors for the *h*ARH3 E41A:H2BS7mar, *h*ARH3 E41A:dimer, *h*ARH3 E41A:α-NAD⁺, and *Lch*ARH3:meADPr structures reported in this paper have been deposited in the RCSB Protein Data Bank (www.rcsb.org) under accession codes 7AKS, 7AKR, 7ARW, and 7AQM, respectively (see 'Methods' and Supplementary Table 2). Earlier deposited structural data used in this study are available in the RCSB Protein Data Bank (www.rcsb.org) under accession codes 2FOZ (*h*ARH3 apo form), 5A7R (PARG:dimer), 6D36 (*h*ARH3:ADPr), 6HGZ (*Lch*ARH3:ADPr) and 6HH3 (*Lch*ARH3:ADP-HPD). The somatic mutation data used in this study are available in the COSMIC database (https://cancer.sanger.ac.uk/cosmic) under accession codes COSM5033871 (D34G), COSM5992851 (T76R), COSM3727906 (S185P), COSM83890 (L186V), and COSM6262998 (G270C) (Supplementary Table 3). Further requests for information, resources and reagents should be directed to and will be fulfilled by the corresponding authors. Source data are provided with this paper.

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

## Acknowledgements

We would like to thank Kerryanne Crawford, Marcin Suskiewicz and Domagoj Baretic for critical comments to the manuscript, Marcin Suskiewicz for assistance with mutant generation, and Diamond Light Source for accessto beamlines I03 and I24 (proposal numbers mx18069 and mx23459), which contributed to the results presented here. Work in I.A.'s laboratory was supported by Wellcome Trust (101794, 210634); Biotechnology and Biological Sciences Research Council (BB/R007195/1); and Cancer Research United Kingdom (C35050/A22284), the A.M. group is funded by German Research Foundation (grants MA4905/4-1, INST38/537-1) and D.A.'s laboratory by the Edward Penley Abraham Research Fund.

## Author contributions

J.G.M.R. performed biochemical experiments, protein crystallisation and data analysis, Q.L. synthesised and validated linear and branched PAR and meADPr, V.Z. and A.A. solved, refined and validated structures, V.Z. carried out structural modelling, J.V. synthesised and validated the H2BS7mar peptide, K.H.E. performed HRMS analysis, J.M.R. and S.C.K. performed branching analysis in cells, J.G.M.R., D.A., GAvdM., J.S.O.M., J.M.R., A.M., D.V.F. and I.A. conceived, designed and coordinated the study, and J.G.M.R. and I.A. wrote the manuscript with input from all authors.

## Competing interests

The authors declare no competing interests.
