## [Peer Review File · Nature Communications]

REVIEWER COMMENTS

Reviewer #1 (Remarks to the Author):

Poly(ADP-ribosylation) is a very complex PTM associated with many pathological processes. The PTM consists of multiple ADP-ribose moieties linked into chains that can number in the hundreds. To make matters even more complex, these chains are also branched.

Although this PTM has been known for many decades, its formation and degradation processes are still poorly understood. This is also due to the lack of many synthetic tools. In the submitted manuscript, the authors address the processes of PAR degradation by PARG and ARH3 using synthetic PAR molecules. For this purpose, a branched PAR residue was synthesized for the first time.

The study of the process of PAR reversal is a very topical and important issue. The synthesis of the PAR molecules presented is challenging and the synthesis alone would be impressive.

Nevertheless, the submitted manuscript has many weaknesses that preclude publication in its present form. These are elaborated below:

Synthetic part:

- 1) It is almost impossible to follow the flow of the synthetic steps, also because the numbering of the intermediates and also the final products in Scheme 1 in the manuscript are completely different from those in the experimental part. This needs to be adjusted.
- 2) Some steps described in the experimental part are nonsensical: Why is the path from 19 to 21 shown (start on page 24 SI)?
- 3) On page 24 it says: "Compound 19 (4.7 g, 5.61 mmol) and 22 (5.17 g, 7.86 mmol) were co-evaporated with toluene". There is no compound 22 in the whole manuscript.
- 4) It seems that two synthesis approaches for the central branch point are presented. Why?
- 5) The procedure for the synthesis of 26 (page 27 SI) should cleave the Adenosine-NH-Bz as well. Why is this not the case?
- 6) The solid phase synthesis remains puzzling. It says that solid phase 31b and solid phase 2a (if the 2a from Scheme 1 is meant) were used. In Cycle B, 3 was then used to obtain the branched PAR. 3 in Scheme 1 of the manuscript is tri-benzoyl-ribose. This certainly does not react with the two solid phases to give the desired product.

7) In line 105 of the manuscript they write of "8 steps, 29% overall yield", which is really impressive for 8 steps. But if you do the math yourself it is only 8%! Also the other stated yield of 28% does not seem to be correct.

At this point I would like to end my points on the descriptions of the synthesis, although I am sure that there are still further inconsistencies. Should the authors submit a revised manuscript, it must a) match the numbering in the manuscript and the SI, b) include a review schematic in the SI that meticulously lists all steps, and c) include all ^1H and ^{13}C NMR spectra as a check for purity. As it stands, this part is impossible to evaluate let alone reproduce.

Biochemical part:

1) Line 128 they write: "...and high-resolution mass spectrometry (HRMS)". I can't find this data or the experimental part for it.

2) Starting with line 161 they write: "and are annotated in the COMIC database for cancer associated mutations, which may indicate an influence on disease progression in some cancer types [<https://cancer.sanger.ac.uk/cosmic>]." First of all: it should be COSMIC. And it would be good to know which mutations are meant.

3) The mechanism of cleavage drawn in 3c above makes no sense to my knowledge of organic chemistry. Cleavage by an E2-like elimination mechanism would only work if the H to be cleaved and the leaving group were trans (antiperiplanar) to each other. This is not the case here - the opposite is true. Also, the enol formed would tautomerize to a ketone. Anyway: no water can be nucleophilically added to an enol or a ketone in the alpha position.

Taken together, I would like to state that due to the insufficiently described synthesis and documentation of the purity of the products, it is also not possible to conclusively assess the biochemistry. This manuscript needs a very thorough revision!

Reviewer #2 (Remarks to the Author):

This is very interesting study that focusses on the impact of PAR structure on its turnover and the catalytic mechanisms underlying the function of ARH3. For the first time, the authors succeed in producing a PAR branch point, which is an important achievement in the field and enables detailed mechanistic studies. The authors show that PARG but not ARH3 can efficiently degrade PAR branch

points, while ARH3 targets linear PAR chains. This has important implications for tuning the duration of PAR signals and defining effector functions of PARylation.

Furthermore, four crystal structures of ARH3, with a PARylated substrate peptide, ADP-ribose, alpha-NAD⁺ and O-methyl-ADP-ribose, offer explanations for the different activities of ARH3 compared to PARG, despite the similar chemical nature of the bonds these enzymes process, and reveal that ARH3 catalysis depends on critical changes in the conformation of a catalytic glutamate and magnesium coordination.

The findings are novel, exciting and relevant to a broad readership studying the wide range of PAR-related cellular processes, including the DNA damage response and signal transduction.

The data are solid and of high quality. However, the clarity of the manuscript could definitely be improved.

Specific comments

Major points

With each of the crystal structures in a different crystal form, can the authors convince the reader that conformational differences in the Glu41 flap are not due to crystal packing?

A figure displaying the solved structures side by side, in the same respective orientations, would be very helpful, at least as a supplementary figure. Likewise, it would be useful to show and compare the structures of the different ligands (especially for the broader readership) and explain more extensively the rationale for choosing these ligands. For example, what is the rationale of determining the structure of LchARH3 with meADPr as a ligand? I am aware that the present study builds on previous work (references 27-29), including that of the authors; however, with a few extended explanations and illustrations, the manuscript could be more self-contained and accessible to a broader readership. It is currently very readable to those familiar with the previous work but not quite as accessible for those who are not.

The cartoon (ribbon) colours in the structural representations are too similar to each other to be clearly distinguished. More distinctively different colours are strongly advised.

p. 6, paragraph 1: The authors state that among the mutated residues, only Glu41 is crucial for ARH3 activity, but Figure 1e shows a number of other mutants that also abolish activity. Can the authors please clarify this?

p.6, paragraph 2: The structural approach and findings could be better described and much more effectively illustrated by figures. For example, it is not immediately clear that the solved structures of human ARH3 have Glu41 mutated. Supplementary Table 1 should also clarify that these are mutants.

There are numerous statements not supported by figures. For example, there is no figure to clearly illustrate the isostructural orientation of the AMP moiety for all the co-crystallised ligands. The peptide orientation, described in the paragraph, is also not obvious in the figure, and it is not possible to appreciate how the peptide stabilises the Glu41 flap in the open state.

p.8, paragraph 1: Thr317 is not shown in Figure 2d and e and is not labelled in panel f. Again, colours representing the different states are difficult to distinguish.

It would really help if the authors could illustrate the "unusual five-coordinated distorted square-pyramidal geometry around [MgII]". This has important implications for the proposed reaction mechanism.

p.8, paragraph 1: The authors propose that the nicotinamide moiety of alpha-NAD⁺ affects MgII positioning and thereby induces flap opening. This proposed mechanism cannot be appreciated from the figures.

p.8, paragraph 2: How is Glu41 "shielded from the aqueous environment by the peptide"? There is no figure to clearly illustrate this observation.

Figure 2: The legend refers to structural elements such a loop L1 and helix alpha2, but these are not labelled. In general, improved labelling would make the figures clearer and more self-explanatory. In particular, panel f needs labelling (side chain, Mg²⁺, which could otherwise be mistaken for water)

Figure 1d: What is the rationale for including HPF1 in the assay?

The authors could spend a few more words how the present study expands the understanding of the ARH3 mechanism beyond the studies referenced as 27-29.

The ARH3:peptide co-crystal structure is particularly exciting. Can the structure provide new insights into the serine specificity of ARH3?

Some readers may argue that modelling E41 into structures that bear an E41A mutation and then deriving mechanistic insights is risky. The authors could do more to assure/convince these readers.

Minor points

p.3, paragraph 2: "Evolutionary"  "Evolutionarily"; "side"  "site"; "substrate-binding"  "substrate binding"

p.6, paragraph 1: "COMIC"  "COSMIC"; "phosphor-ribose"  "phospho-ribose"

p.7, end of paragraph 1: Is "conservation" the best word here? Would "retention" work better?

Figure 1: Panel a only shows the branch point, so the "trimer label" can be removed for clarity. The size of the panels could be increased. More generous labelling could make the figure more self-explanatory. For sample, panel b could include the different conditions instead of Roman numbers referring to the legend. Panel e could be more extensively explained in the legend (what is the assay?). Also in panel e, the labels almost overlap, and arrowheads are barely visible.

Reviewer #3 (Remarks to the Author):

The manuscript by Rack & Liu et al. describes two technically sound but reluctantly related works. One is about the de novo chemical synthesis of branched point tri-ADPr, which is very valuable for structural and biochemical analysis of PAR-regulated biological processes. Using this tri-ADPr, authors clearly demonstrated that, consistent with previous studies, PARG, but not ARH3, is responsible for PAR branch point cleavage. It is pity that authors did not reveal how PARG recognizes and cleaves PAR branch points, which is of great interest for many researchers in the field. Instead, authors studied how ARH3 works in the second part of this manuscript, a topic which has been studied quite extensively. In my opinion, this manuscript is more suitable for two separate papers in specialized journals.

Specific points:

- 1) In figures 2/3, a lack of overall structures made it hard to understand the zoomed-in structural details.
- 2) Comparison of structures with previous reported ARH3 apo/ADPr bound structures should be added. Glu41 interaction with MgII should be compared with Apo hARH3 structure (PDB: 2FOZ).
- 3) Figure 1D, it looks like there is fewer PARP1 for G972R set compared to the other sets. How about Y986S? The Y986S result should be discussed. It looks like as efficient as G972R. It would be better if the result was quantified.
- 4) Is it possible to get the complex structure of meADPr with Hs ARH3, instead of Lch ARH3 for direct comparison?
- 5) Line 133-136, is it possible that the observed cleavage of ribose(1'''→2'')ribose as the first step for PARG clearance of the branching point resulted from the terminal meADPr unit of synthesized branching trimer? Please discuss.
- 6) Line 185-187, figure should be provided for the peptide binding groove.
- 7) Line 274-278, "facing away from the peptide active site" is hard to see from the provided model figure. One would imagine the flexibility of the substrate peptides of Asp-/Glu-linkages, especially the Glu-linkage one with longer side chains.
- 8) Line 25, typo, it should be S-glycosidic
- 9) Line 32, 36, 136, the ribose junction for the branching point should be ribose(1'''→2'')ribose.
- 10) Line 36, what does "resolution" mean here?
- 11) Scheme 1. Product (I) there is a Bz modification on the adenine ring.
- 12) Line 153-155, please clarify that E41A mutant was used for crystallization.
- 13) Supplementary Fig. 2c, lower panel showed simulated result for me-ADPr instead of ADPr.

We are grateful to the reviewers and editors for their constructive comments. Here is our point-by-point response.

Reviewer #1 (Remarks to the Author):

Poly(ADP-ribosylation) is a very complex PTM associated with many pathological processes. The PTM consists of multiple ADP-ribose moieties linked into chains that can number in the hundreds. To make matters even more complex, these chains are also branched.

Although this PTM has been known for many decades, its formation and degradation processes are still poorly understood. This is also due to the lack of many synthetic tools. In the submitted manuscript, the authors address the processes of PAR degradation by PARG and ARH3 using synthetic PAR molecules. For this purpose, a branched PAR residue was synthesized for the first time.

The study of the process of PAR reversal is a very topical and important issue. The synthesis of the PAR molecules presented is challenging and the synthesis alone would be impressive.

Nevertheless, the submitted manuscript has many weaknesses that preclude publication in its present form. These are elaborated below:

Synthetic part:

1) It is almost impossible to follow the flow of the synthetic steps, also because the numbering of the intermediates and also the final products in Scheme 1 in the manuscript are completely different from those in the experimental part. This needs to be adjusted.

We regret the unfortunate numbering mistakes in the manuscripts. We corrected the numbering and thoroughly reworked the synthetic part of the paper to facilitate the reading. We are grateful to the reviewer for the thoughtful remarks.

2) Some steps described in the experimental part are nonsensical: Why is the path from 19 to 21 shown (start on page 24 SI)?

At the beginning of our synthesis, we use tri-PMB donor **4** to couple with **3** to get disaccharide **20** in high yield. Similarly, we first chose to use the same donor to get trisaccharide **23**, however the deprotection step proved troublesome and gave a low yield (route A, 47%). Then we switched to another donor **5** and finally get **25** in much improved yield (79%, route B) which is beneficial for big scale synthesis. We feel that the route A should be mentioned for its synthetic merits (only one and simple donor is needed). In the revised paper we have changed the schemes to emphasise the optimised approach.

3) On page 24 it says: "Compound 19 (4.7 g, 5.61 mmol) and 22 (5.17 g, 7.86 mmol) were co-evaporated with toluene". There is no compound 22 in the whole manuscript.

This is an unfortunate numbering mistake, which we regret. We corrected the numbering. As said, we are grateful to the reviewer for the help with the correcting of the synthetic part of the paper.

4) It seems that two synthesis approaches for the central branch point are presented. Why?

The explanation is given in the section 2 above.

5) The procedure for the synthesis of 26 (page 27 SI) should cleave the Adenosine-NH-Bz as well. Why is this not the case?

Benzamide protection at the exocyclic amine that is sensitive to a prolonged treatment with ammonia under ambient conditions (16 h, room temperature) is known to be resistant to a short treatment with alkali at low temperature (2 h at 0 °C), while the esters are saponified with this relatively mild method. Such fortunate behaviour of the N-Bz-protection of adenine base has been exploited on a number of occasions. We added the relevant references to the revised paper. The explanation why the N-Bz protection survives here could be the well-established fact that the amides are intrinsically more stable to basic hydrolysis than esters due to lower electrophilicity of carbonyl attached to nitrogen as compared to one connected to an oxygen atom. A more sophisticated, although still speculative, explanation can follow from the "hard" nature of hydroxide, which acts primarily as a base and deprotonates the secondary amide protecting it from the subsequent nucleophilic attack. In contrast, "softer" and less basic ammonia, which is unable to deprotonate the amide functionality, effectuates the deprotection of N-Benzoyl by the direct nucleophilic attack on the carbonyl of the amide.

6) The solid phase synthesis remains puzzling. It says that solid phase 31b and solid phase 2a (if the 2a from Scheme 1 is meant) were used. In Cycle B, 3 was then used to obtain the branched PAR. 3 in Scheme 1 of the manuscript is tri-benzoyl-ribose. This certainly does not react with the two solid phases to give the desired product.

This is again a regrettable numbering mistake, which caused this confusion. We meant the phosphoramidite 9 (it is compound 4 in the revised paper). We corrected all numbering and wrong "pointers" throughout the manuscript.

7) In line 105 of the manuscript they write of "8 steps, 29% overall yield", which is really impressive for 8 steps. But if you do the math yourself it is only 8%! Also the other stated yield of 28% does not seem to be correct.

We have to apologise here again for the lack of clarity. The overall yield from 3 to 6 (old numbering, 6 to 14 in the revised paper) is indeed 29% overall if it is calculated over the optimal route (route B in the original submission). It is humbling to hear that the reviewer finds this result impressive. The misstated yield of 28% in the main text should be indeed 24% as in the Scheme. We are grateful to the reviewer for spotting this mistake. We removed as poorly informative all statements on the overall yields from the revised paper. We hope it is an appropriate decision.

At this point I would like to end my points on the descriptions of the synthesis, although I am sure that there are still further inconsistencies. Should the authors submit a revised manuscript, it must a) match the numbering in the manuscript and the SI, b) include a review schematic in the SI that meticulously lists all steps, and c) include all ¹H and ¹³C NMR spectra as a check for purity. As it stands, this part is impossible to evaluate let alone reproduce.

We thoroughly reworked the synthetic part of the paper, corrected all numbering mistakes, removed inconsistencies and expanded the Schemes to include all synthetic steps as requested. We profusely apologise to the referee for the confusion we inadvertently caused. We thank the reviewer for the helpful remarks. We included the copies of ¹H, ¹³C and ³¹P NMR spectra of all compounds into the Supplementary Materials following the description of the synthesis. The spectra were originally uploaded as Supplementary Dataset and we hope that this change in presentation will make them more accessible for the reader.

Biochemical part:

1) Line 128 they write: "...and high-resolution mass spectrometry (HRMS)". I can't find this data or the experimental part for it.

We apologise to the referee if this was not clear. The experimental data was described as liquid chromatography-mass spectrometry (LC-MS). This has now been corrected as liquid chromatography-high resolution mass spectrometry (LC-HRMS).

2) Starting with line 161 they write: "and are annotated in the COMIC database for cancer associated mutations, which may indicate an influence on disease progression in some cancer types [<https://cancer.sanger.ac.uk/cosmic>]." First of all: it should be COSMIC. And it would be good to know which mutations are meant.

We thank the reviewer for spotting the typo and agree with the suggestion of highlighting the mutations derived from the literature and database. We have altered Fig. 2 to highlight the mutation derived from the "COSMIC" database. In addition, we added further information about the COSMIC mutations to Supplementary Tab. 3. We hope that this will make the data more accessible for the reader.

3) The mechanism of cleavage drawn in 3c above makes no sense to my knowledge of organic chemistry. Cleavage by an E2-like elimination mechanism would only work if the H to be cleaved and the leaving group were trans (antiperiplanar) to each other. This is not the case here - the opposite is true. Also, the enol formed would tautomerize to a ketone. Anyway: no water can be nucleophilically added to an enol or a ketone in the alpha position.

We appreciate the concern regarding the E2-like elimination mechanism. We, however, maintain that the "eliminative" mechanism does make certain sense here. The H that is abstracted is antiperiplanar to the leaving group, which is the aglycon in the alpha-configuration (e.g. serine side chain). If the enol, which is the product of this elimination, is protonated at the C-2 (ribose numbering) before it tautomerises to the C-2 ketone, the resulting oxocarbenium ion would constitute a "bona fide" electrophilic intermediate in the addition of water to the anomeric position. We do agree, however, that this "lyase-like" mechanism is somewhat far-fetched and therefore we removed it from the revised paper. In addition, to gain further insights into the reaction progression following ARH3:substrate complex formation, we performed reaction in the presence of increasing amounts of methanol. Methanol is known as a nucleophile able to react with oxocarbenium intermediates in enzymatic reaction (methanolysis), thus detection of methyl-ADPr is a strong indicator of this intermediate within the reaction. Indeed, we could detect ARH3-dependent production of meADPr in a concentration-dependent manner, thus confirming that reaction progression via an oxocarbenium intermediate is the most likely mechanism. We amended the reaction mechanism to reflect this finding.

Taken together, I would like to state that due to the insufficiently described synthesis and documentation of the purity of the products, it is also not possible to conclusively assess the biochemistry. This manuscript needs a very thorough revision!

Reviewer #2 (Remarks to the Author):

This is very interesting study that focusses on the impact of PAR structure on its turnover and the catalytic mechanisms underlying the function of ARH3. For the first time, the authors succeed in producing a PAR branch point, which is an important achievement in the field and enables detailed mechanistic studies. The authors show that PARG but not ARH3 can efficiently degrade PAR branch points, while ARH3 targets linear PAR chains. This has important implications for tuning the duration of PAR signals and defining effector functions of PARylation.

Furthermore, four crystal structures of ARH3, with a PARylated substrate peptide, ADP-ribose, alpha-NAD⁺ and O-methyl-ADP-ribose, offer explanations for the different activities of ARH3 compared to PARG, despite the similar chemical nature of the bonds these enzymes process, and reveal that ARH3 catalysis depends on critical changes in the conformation of a catalytic glutamate and magnesium coordination.

The findings are novel, exciting and relevant to a broad readership studying the wide range of PAR-related cellular processes, including the DNA damage response and signal transduction.

The data are solid and of high quality. However, the clarity of the manuscript could definitely be improved.

Specific comments

Major points

With each of the crystal structures in a different crystal form, can the authors convince the reader that conformational differences in the Glu41 flap are not due to crystal packing?

We acknowledge that this can be a concern for the reader and have added a Supplementary Fig. 9 showing the higher order packing of the different structures. Despite the different space groups the major packing features of the three human ARH3 structures are comparable, and all structures have the Glu41-flap facing a solvent channel, thus allowing flexibility within the crystal packing and not forcing a specific conformation. This is further supported by generally higher B-factors in this region of ARH3 together with some missing density for small parts of the Glu41 flap surrounding helix α_2 (see e.g. Fig.3a), which indicates a high degree of flexibility within this region of the molecule. We also included comparison between the different chains within each reported structure (Supplementary Fig. 8) and show that even in cases where packing contacts were identified these have no influence on the overall packing as the Glu41-flap conformations are comparable to those in chains lacking these contacts. We hope that this addition dispels the possible doubts of the readers that our observed conformational changes in the Glu41-flap are induced by crystal packing.

A figure displaying the solved structures side by side, in the same respective orientations, would be very helpful, at least as a supplementary figure.

We included Supplementary Fig. 6 showing an overall structural comparison between our here reported structures and the earlier solved LchARH3:ADPr and hARH3:ADPr complexes.

Likewise, it would be useful to show and compare the structures of the different ligands (especially for the broader readership) and explain more extensively the rationale for

choosing these ligands. For example, what is the rationale of determining the structure of LchARH3 with meADPr as a ligand?

We included Supplementary Tab. 1 showing a comparison between the different ligands used in this study and included more explanatory text. Unfortunately, we were unable to produce hARH3:meADPr crystals of high enough quality to report here. We chose to utilise our earlier established LchARH3 soakable crystal system as an alternative due to limitations placed on us by the current COVID-19 pandemic. Our earlier experience with LchARH3 showed that ligand overall structure and ligand interactions are very similar to hARH3 and that findings in either can be transferred across species. We hope that the inclusion of Figures (e.g. Supplementary Fig. 6) comparing the structures and ligand placement will help the reader to appreciate this high degree of similarity.

I am aware that the present study builds on previous work (references 27-29), including that of the authors; however, with a few extended explanations and illustrations, the manuscript could be more self-contained and accessible to a broader readership. It is currently very readable to those familiar with the previous work but not quite as accessible for those who are not.

We appreciate that this work was written with a specialist in mind and have adjusted explanation throughout the manuscript to help readers less familiar with previous work on ARH3. In addition we included several figures highlighting structural features previously not shown, incl. overall structures, binding surfaces, magnesium coordination detail, etc. We hope that these changes make this a stand-alone work accessible to all interested readers.

The cartoon (ribbon) colours in the structural representations are too similar to each other to be clearly distinguished. More distinctively different colours are strongly advised.

We change the colour scheme of our structural representations and hope that this will make the figures more accessible for the reader.

p. 6, paragraph 1: The authors state that among the mutated residues, only Glu41 is crucial for ARH3 activity, but Figure 1e shows a number of other mutants that also abolish activity. Can the authors please clarify this?

We apologise for the imprecision of the statement. What was meant is that E41A is the only mutant that abolishes activity without affecting magnesium or ligand binding and is therefore ideal for our structural work. The other mutants affecting activity are all equally important for ADP-ribose moiety or metal coordination and therefore not suitable for ligand structure crystallisation. We altered the text to better reflect this fact and the rationale for showing the other mutants.

p.6, paragraph 2: The structural approach and findings could be better described and much more effectively illustrated by figures. For example, it is not immediately clear that the solved structures of human ARH3 have Glu41 mutated. Supplementary Table 1 should also clarify that these are mutants.

We agree with the reviewer and re-phrased the text to highlight clearly that the hARH3 variant used was the E41A mutant (the text states now: "Therefore, we chose to crystallise the hARH3 E41A substrate complexes as this mutation prevents substrate cleavage during crystal formation while retaining a functionally relevant magnesium coordination."). In

addition, we updated the crystallographic table to clarify that the hARH3 structures are mutants.

There are numerous statements not supported by figures. For example, there is no figure to clearly illustrate the isostructural orientation of the AMP moiety for all the co-crystallised ligands. The peptide orientation, described in the paragraph, is also not obvious in the figure, and it is not possible to appreciate how the peptide stabilises the Glu41 flap in the open state.

We added additional figures and explanation to better illustrate our findings and hope that these will aid in increasing readability.

p.8, paragraph 1: Thr317 is not shown in Figure 2d and e and is not labelled in panel f. Again, colours representing the different states are difficult to distinguish.

We added the missing Thr317 residue, added labels and change the colour scheme. Together we hope that this improves the clarity of the figure.

It would really help if the authors could illustrate the "unusual five-coordinated distorted square-pyramidal geometry around [MgII]". This has important implications for the proposed reaction mechanism.

We added new figures (Fig. 2d+e and Supplementary Fig. 12) detailing the magnesium coordination in all our structures as well as relevant previously solved structures (apo/ADPr bound of hARH3 and LchARH3) for comparison. We hope these illustrate better the differences in coordination geometry at the Mg_{II} centre.

p.8, paragraph 1: The authors propose that the nicotinamide moiety of alpha-NAD⁺ affects MgII positioning and thereby induces flap opening. This proposed mechanism cannot be appreciated from the figures.

We apologise if the explanation was unclear. Alpha-NAD⁺, in contrast to the H2BS7mar structure, does not affect the positioning of the MgII ion, but rather sterically displaces the axial water molecule. However, the nicotinamide moiety – most likely due to its positive charge – is positioned at a relatively long distance from the MgII ion, thus not acting as a ligand for the MgII ion. This leads to an under-satisfied first coordination sphere, which together with the ring slipping-like orientation of the nicotinamide ring represent an activated substrate bond state. We re-wrote the text to reflect this fact and hope that together with the new coordination figure (Supplementary Fig. 12 and Tab. 5) this will make it clear to the reader why alpha-NAD⁺ is a good substrate despite the fact that it does not induce the same coordination changes as a serine-ADP-ribosylated peptide.

p.8, paragraph 2: How is Glu41 "shielded from the aqueous environment by the peptide"? There is no figure to clearly illustrate this observation.

We added a figure to show this finding (Supplementary Fig. 13).

Figure 2: The legend refers to structural elements such a loop L1 and helix alpha2, but these are not labelled. In general, improved labelling would make the figures clearer and more self-explanatory. In particular, panel f needs labelling (side chain, Mg²⁺, which could otherwise be mistaken for water)

We improved the labelling of our figures and hope this increased their clarity.

Figure 1d: What is the rationale for including HPF1 in the assay?

We conducted all experiments in the presence of HPF1 since PARP1 requires HPF1 to produce Ser-ADP-ribosylation on sites that are DNA damage response-relevant. Exclusion of HPF1 would lead to the modification of glutamate residues – a modification that cannot be reversed by ARH3. We added text to highlight this fact and explain our rationale for adding HPF1 to our assays.

The authors could spend a few more words how the present study expands the understanding of the ARH3 mechanism beyond the studies referenced as 27-29.

Thank you for the suggestion. We added some more text to make the manuscript more self-contained.

The ARH3:peptide co-crystal structure is particularly exciting. Can the structure provide new insights into the serine specificity of ARH3?

Yes – the data highlight a convincing mechanism for the efficient cleavage of Ser-ADP-ribosylation and show why PAR degradation is considerably slower. In terms of specificity, the positioning of the peptide also suggest Glu-ADP-ribosylated peptides cannot be accommodated in the same orientation due to the coordination required for cleavage of the O-glycosidic ester bond (modelled for OAADPr). We added these insights to the discussion section.

Some readers may argue that modelling E41 into structures that bear an E41A mutation and then deriving mechanistic insights is risky. The authors could do more to assure/convince these readers.

We acknowledge that this could be of concern to some reader. However, this work as well as previous work by others and us clearly establishes Glu41 as a catalytically crucial residue. Therefore, we would argue that inclusion into the mechanism is prudent rather than risky. Regarding our modelling efforts: The reintroduction of a mutated side chain would normally not be expected to force major conformational changes. In addition, previous hARH3:ADPr structures show flexibility in the Glu41 arrangement and our model is well in line with these structures. Further, the bound H2BS7mar present in our models places major steric constraints on the Glu41 placement with the presented Glu41 rotamer being one of only 2 possible favourable rotamers. The second is facing away from the catalytic centre and if placed participation in the reaction cannot obviously explained from the structure. Lastly, we observed the presence of a water molecule in place of this favourable rotamer within the solved structure, thus supporting the placement of a polar group in this position. In conclusion, we would argue that the limitation in rotamer selection, shielding by the bound

peptide, presence of the water molecule and participation in the catalytic mechanism in combination support our model. One should also note the high resolution of our structures which makes the considerations of geometry and rotamers more confident than would be the case for medium-resolution models.

To assure the reader we included Supplementary Fig. 13 showing the steric restrictions as well as added more explanatory text.

Minor points

p.3, paragraph 2: "Evolutionary"  "Evolutionarily"; "side"  "site"; "substrate-binding"  "substrate binding"

We made the changes.

p.6, paragraph 1: "COMIC"  "COSMIC"; "phosphor-ribose"  "phospho-ribose"

We made the changes.

p.7, end of paragraph 1: Is "conservation" the best word here? Would "retention" work better?

We made the changes.

Figure 1: Panel a only shows the branch point, so the "trimer label" can be removed for clarity. The size of the panels could be increased. More generous labelling could make the figure more self-explanatory. For sample, panel b could include the different conditions instead of Roman numbers referring to the legend. Panel e could be more extensively explained in the legend (what is the assay?). Also in panel e, the labels almost overlap, and arrowheads are barely visible.

We made the changes.

Reviewer #3 (Remarks to the Author):

The manuscript by Rack & Liu et al. describes two technically sound but reluctantly related works. One is about the de novo chemical synthesis of branched point tri-ADPr, which is very valuable for structural and biochemical analysis of PAR-regulated biological processes. Using this tri-ADPr, authors clearly demonstrated that, consistent with previous studies, PARG, but not ARH3, is responsible for PAR branch point cleavage.

The role of ARH3 for branching has so far never been addressed in vivo. We added the new result (Fig 2d) demonstrating that ARH3 indeed does not have a significant effect on branching using a recently established U2OS ARH3^{-/-} cell culture model. This experiment also provides additional connection between two parts of the paper.

It is pity that authors did not reveal how PARG recognizes and cleaves PAR branch points, which is of great interest for many researchers in the field. Instead, authors studied how ARH3 works in the second part of this manuscript, a topic which has been studied quite extensively. In my opinion, this manuscript is more suitable for two separate papers in specialized journals.

We agree that the recognition of PAR branch points by PARG is of great interest to the field. While a detailed analysis is outside the scope of this manuscript, we took up this valuable comment and performed modelling of the branch point molecule into the previously solved structure of PARG with a PAR dimer. The results show that insertion of ADPr with the double-linkage within the active site (aka the linear continuation ADPr) sterically hinders the catalytic Glu756 to interact with the 2' O of the ADPr. This hindrance is absent in model placing the first branched ADPr in the active site. While only giving a static image of the PARG:branch point interaction, these findings support our observations and we hope that they will stimulate further work in the field.

Specific points:

1) In figures 2/3, a lack of overall structures made it hard to understand the zoomed-in structural details.

We added overall structures (Supplementary Fig. 6) to give the reader a better orientation with respect to the details shown. We hope this helps to improve the clarity of the presentation.

2) Comparison of structures with previous reported ARH3 apo/ADPr bound structures should be added. Glu41 interaction with MgII should be compared with Apo hARH3 structure (PDB: 2FOZ).

We appreciate the point and added the suggested comparison.

3) Figure 1D, it looks like there is fewer PARP1 for G972R set compared to the other sets. How about Y986S? The Y986S result should be discussed. It looks like as efficient as G972R. It would be better if the result was quantified.

We appreciate the comment and altered our description of the results: according to Aberle et al. (NAR, 2020) and Rolli et al. (Biochemistry, 1997), as well as our own observations, all three mutations have reduced activity compared to wild type with Y986S showing near wt, G972R reduced and Y986H increased branch frequency. The inherent differences in chain length generation make precise quantification and comparisons difficult. Therefore, we chose to maintain a qualitative approach and focus on the clear differences. We acknowledge that our initial description of the results could be seen as asserting a difference in degradation between Y986S and G972R and therefore we reformulated the description and state now that "[...]the stability of PARP1 Y986H-derived polymers was increased in comparison to G972R- or Y986S-derived PAR (Fig. 2e), thus further supporting our finding that ARH3 hydrolysis of PAR is slowed by the presence of branch points."

4) Is it possible to get the complex structure of meADPr with Hs ARH3, instead of Lch ARH3 for direct comparison?

We have tried the human orthologue but our initial attempts have been unsuccessful. Due to limitation placed on us by the current COVID-19 pandemic, we decided therefore to pursue a

soaking approach with our previously established LchARH3 crystal system (Rack et al., Cell Chem Biol, 2018). Next to the structural observation we presented biochemical and biophysical evidence that hARH3 and LchARH3 show near identical ligand binding and assay behaviour and that results obtained in one species can generally be transferred into the other. Furthermore, the observations relevant to this manuscript concern the absolutely conserved magnesium centre, therefore we believe LchARH3 to be a suitable model system for the observation. To make this argument even more persuasive we added further comparison structure – also of LchARH3:ADPr – into the manuscript and hope the reader is now in the position to judge our conclusions (Supplementary Fig. 6, 8, 10, 12).

5) Line 133-136, is it possible that the observed cleavage of ribose(1''→2'')ribose as the first step for PARG clearance of the branching point resulted from the terminal meADPr unit of synthesized branching trimer? Please discuss.

Observation of an ADPr dimer as a reaction intermediate can only be explained if either the linear extension or branched ADPr – and not the meADPr – is placed within the active site. Specifically, our observation would require the branched ADPr group to be within the active site (see also new Supplementary Fig. 3). While we cannot fully exclude a long range influence of the methyl group on the reaction, we do not believe this to be the case as the part of the ligand containing the methyl moiety cannot be observed in the earlier reported crystal structure of PARG in complex with a linear PAR dimer (PDB 5A7R; Supplementary Fig. 3). The situation in this crystal appears to be comparable to our ARH3:dimer structure, which shows increased flexibility in the distal ribose resulting in the absence of electron density for this part of the molecule (which would contain the methyl group). Furthermore, in the paper reporting the PARG:dimer structure (Lambrecht, J Am Chem Soc, 2015) the authors tested the activity of PARG against the ADPr dimer, an AMP-ADPr dimer and 1''-propargyl-ADPr-ADPr and found no difference in PARG activity, thus strongly suggesting that addition of modification at the terminal 1'' position does not influence the reaction. These earlier observations together with our new structural models, thus strongly suggest that selectivity is a result of steric constraints within the active site imposed by the branch point itself.

6) Line 185-187, figure should be provided for the peptide binding groove.

We added Supplementary Fig. 10a to illustrate the limited peptide:ARH3 interaction.

7) Line 274-278, “facing away from the peptide active site” is hard to see from the provided model figure. One would imagine the flexibility of the substrate peptides of Asp-/Glu-linkages, especially the Glu-linkage one with longer side chains.

We updated the figure (now Supplementary Fig. 14) and hope this will improve clarity.

8) Line 25, typo, it should be S-glycosidic

Thank you for spotting the typo. We corrected it.

9) Line 32, 36, 136, the ribose junction for the branching point should be ribose(1''→2'')ribose.

If the reviewers agrees, we would like to keep this nomenclature which we believe is not incorrect: within the ADP-ribose molecule we denote positions within the proximal ribose as ' (prime) and in the distal ribose as '' (prime-prime). Therefore, introducing triple-prime would suggest the presence of an additional ribose in one of the subunits of the PAR molecule. To specifically describe the branch point situation, rather than a specific bond, in which 3 riboses are conjugated we utilise the ribose(1''→2'')ribose(1''→2')ribose annotation denoting that the middle molecule is both acceptor and donor. To further improve clarity, we propose a new nomenclature to denote specific ADPr units within a polymer. As we see our work as a step towards the study of more complex, well-defined PAR molecules and we hope this will encourage discussion within the ADP-ribosylation community to develop a common language to describe these molecules.

10) Line 36, what does “resolution” mean here?

We alter the phrasing for clarity and substituted “resolution” with “hydrolysis”

11) Scheme 1. Product (I) there is a Bz modification on the adenine ring.

We apologise for the mislabelling in the scheme and corrected the error in the update synthesis description and associated schemes and figure.

12) Line 153-155, please clarify that E41A mutant was used for crystallization.

We clarified that all human ARH3 structures were obtained using the Glu41Ala mutant.

13) Supplementary Fig. 2c, lower panel showed simulated result for me-ADPr instead of ADPr.

We apologise for the oversight and corrected the panel.

REVIEWERS' COMMENTS

Reviewer #1 (Remarks to the Author):

The authors have addressed the issues raised by me (mainly concerning the synthesis) with great care. As already mentioned in my earlier report, the synthesis is quite an achievement and now has the appropriate coverage in the manuscript.

Some minor issues:

The authors write (line 184) „...PARG catalytic mutant E756N...”. Should this read “catalytically impaired mutant” or alike?

The authors write (line 217) “...ether O-glycosidic bond...” which is in fact an “acetal O-glycosidic bond”

The authors write (line 441) “serine carbonyl” is this the one of the amino acid backbone? The side chain does not possess a carbonyl (it has an alcohol functionality)

Taking together, after fixing these really minor issues I enthusiastically recommend publication of this manuscript.

Reviewer #2 (Remarks to the Author):

I thank the authors for the extensive revisions. All points I raised during the first round of review are addressed. There only remain a few relatively small issues, which can be addressed before publication.

Main points

In the beginning of the Results section, the authors state that "chain branching has come into focus as a key determinant of the cellular consequences of PAR signalling." It is far from established that branching is a "key determinant", and I suggest to re-word to state that it is a potential determinant.

Supplementary Figure 3: It would help to label the 2" and 3" OH groups.

I am supportive of the proposed nomenclature to describe PAR chains (p. 7).

I realise I did not raise this point earlier, but including a quantification of Figure 2e (with repeats) would be helpful, if these data are available.

Minor points

abstract: revealed [the] molecular basis

p. 2, paragraph 1: and [the] DNA damage response

p. 2, paragraph 2: [and] hence terminates

p. 3, end: "our finding [suggest a] novel hydrolase mechanism"

Figure 2, legend: Panel d is mistakenly named e.

p. 6, end: "with earlier observation[s]"

Supplementary Figure 3a, legend: "Astericks"  "Asterisks"

p. 8: [Supplementary] Table 1

p. 12, Discussion: "The dynamics of PAR signal turnover [are] a crucial determinant ..."

Reviewer #3 (Remarks to the Author):

Authors have largely addressed technical issues I raised.

We would like to thank the reviewers for their kind comments and their support for the publication of our work. We addressed all the reviewers' comments as detailed below.

Reviewer #1 (Remarks to the Author):

The authors have addressed the issues raised by me (mainly concerning the synthesis) with great care. As already mentioned in my earlier report, the synthesis is quite an achievement and now has the appropriate coverage in the manuscript.

Some minor issues:

The authors write (line 184) „...PARG catalytic mutant E756N...”. Should this read “catalytically impaired mutant” or alike?

We agree that “catalytically impaired” better describes the mutant in this context and changed the wording as suggested.

The authors write (line 217) “...ether O-glycosidic bond...” which is in fact an “acetal O-glycosidic bond”

We corrected the error.

The authors write (line 441) “serine carbonyl” is this the one of the amino acid backbone? The side chain does not possess a carbonyl (it has an alcohol functionality)

Yes, the sentence refers to the backbone carbonyl and we altered the text to read “serine backbone carbonyl” to clarify this fact.

Taking together, after fixing these really minor issues I enthusiastically recommend publication of this manuscript.

Reviewer #2 (Remarks to the Author):

I thank the authors for the extensive revisions. All points I raised during the first round of review are addressed. There only remain a few relatively small issues, which can be addressed before publication.

Main points

In the beginning of the Results section, the authors state that "chain branching has come into focus as a key determinant of the cellular consequences of PAR signalling." It is far from established that branching is a "key determinant", and I suggest to re-word to state that it is a potential determinant.

We altered the sentence to read “Recent studies suggest that the PAR branching frequency is an important determinant of the cellular outcomes of ADP-ribosyl signalling”

Supplementary Figure 3: It would help to label the 2" and 3" OH groups.

We added labels to the figure to indicate these groups.

I am supportive of the proposed nomenclature to describe PAR chains (p. 7).

I realise I did not raise this point earlier, but including a quantification of Figure 2e (with repeats) would be helpful, if these data are available.

We agree in principle that quantification would be desirable. However, the inherent differences in chain length generation make precise quantification and comparisons difficult. Therefore, we chose to maintain a qualitative approach and focus on the clear differences between the stabilisation observed in Y986H and the faster hydrolysis detected in 986S and G972R.

Minor points

abstract: revealed [the] molecular basis

corrected

p. 2, paragraph 1: and [the] DNA damage response

corrected

p. 2, paragraph 2: [and] hence terminates

corrected

p. 3, end: "our finding [suggest a] novel hydrolase mechanism"

corrected

Figure 2, legend: Panel d is mistakenly named e.

corrected

p. 6, end: "with earlier observation[s]"

corrected

Supplementary Figure 3a, legend: "Astericks"  "Asterisks"

corrected

p. 8: [Supplementary] Table 1

corrected

p. 12, Discussion: "The dynamics of PAR signal turnover [are] a crucial determinant ..."

corrected

Reviewer #3 (Remarks to the Author):

Authors have largely addressed technical issues I raised.